# EFFICIENT PARAMETER–SPACE INTEGRATED GRADIENTS FOR DEEP NETWORK OPTIMIZATION

## ABSTRACT

We explore previously unreported properties and practical uses of integrated gradients for training deep neural networks, primarily convolutional models, in the sense of averaging gradients over a continuous range of parameter values at each update step rather than relying solely on the instantaneous gradient. Our contributions are: (a) We show that, across multiple architectures, integrated gradients yield up to 53.5% greater reduction in per-batch loss compared to baseline optimizers. (b) We demonstrate that, for a fixed batch and models prone to ill–conditioned curvature, a single step can approximate more than four predicted updates. (c) We introduce an efficient approximation for ResNet–152 fine–tuning that integrates gradients over hundreds of past training iterations on a fixed batch at each parameter update. This variant is faster per step and easier to parallelize than a single step of a competitive Sharpness–Aware Minimization method, with only moderate memory overhead.
We validate the approach with first-order optimizers (RMSProp, Adam) and a second-order method (SOAP), showing consistent gains across settings. These results suggest that integrated gradients are a promising new direction for improving the generalization and potentially the test–time adaptation of deep models.

## 1 INTRODUCTION

Attribution methods known as integrated gradients (Sundararajan et al., 2017) operate in input and feature space, but their potential for optimization remains largely unexplored. We study parameter–space gradient integration for training: at each step, our approach replaces the instantaneous gradient with an averaged (integrated) gradient (Anonymized, 2025), as formally defined in Appendix A, yielding a drop-in modification portable to both first-order and second-order optimizers.

Building on our earlier results (Anonymized, 2025), we broaden the analysis and applications of averaged gradients. Our contributions are: (a) **Batch-local efficiency:** Significant improvements in *batch*-loss minimization for first- and second-order optimization, enabling rapid test-time adaptation on a fixed batch. (b) **Generalization:** On ResNet-152 fine-tuning, generalization improves with lower per-step compute and simpler parallelization than Sharpness–Aware Minimization (SAM) (Foret et al., 2021), at the expense of additional memory; most competing techniques are not designed to integrate cleanly with modern second-order optimizers (Vyas et al., 2024; Zhang et al., 2023; Gupta et al., 2018). (c) **Multi-update approximation:** Despite prior math foundations (Anonymized, 2025), we show a single step of our algorithm can approximate multiple predicted updates at a cost close to two standard updates, and we introduce a more stable, faster variant that averages gradients for the current batch over hundreds of recent training iterations. (d) **Activation-level impact:** Experiments indicate that averaging at the activation level is particularly important for training efficiency, with theoretical support. (e) **Depth-related pathologies:** Our method targets depth-induced degeneracies ("singularity problems") (Oyedotun et al., 2021). (f) **Applications:** We outline additional use cases in Section 5.

As our approach necessitates modifying the backpropagation procedure, existing implementations of various architectures could not be directly reused. Therefore, we focus on a limited set of models primarily based on convolutional layers, as detailed in Section 6.

## 2 RELATED WORK

**Momentum (EMA of gradients).** Momentum (Liu et al., 2020) maintains an exponential moving average of past gradients across successive iterations (typically over different mini-batches) and can accelerate convergence while reducing oscillations in ill-conditioned directions. In contrast, our approach averages the gradient for a fixed batch along a continuous path in parameter space; momentum aggregates gradients across distinct batches rather than along a parameter trajectory.

**Polyak–Ruppert averaging (parameter iterates).** Averaging model parameters over many updates (Polyak–Ruppert) can stabilize training and improve generalization in certain settings (Ruppert, 1988; Polyak & Juditsky, 1992; Merity et al., 2017; Li et al., 2023; Sun et al., 2010). This procedure is orthogonal to update computation and can be layered atop any optimizer. Our work instead targets the update itself by modifying backpropagation to approximate a parameter-path averaged gradient.

**Perturbated Unit Gradient Descent (PUGD).** Let $g(\theta) = \nabla_\theta \ell(\theta)$ denote the gradient of the loss $\ell$ for the current batch. PUGD computes the unit-normalized sum of $g(\theta)$ and $g(\theta + |\theta| \odot g(\theta))$, where the perturbation is applied elementwise (Tseng et al., 2022). Unlike PUGD, our method averages the gradient along a broader, continuous path in parameter space using a distinct approximation.

**Two–gradient tuning strategies.** Several techniques modulate updates using exactly the two most recent gradients and are implemented as modifications to Adam-style optimizers (Kingma & Ba, 2014). DiffGrad attenuates parameter updates when the current and previous gradients are similar in magnitude (Dubey et al., 2019). SigSignGrad computes an elementwise scaling factor from the sign agreement between the last two gradients, mapping it through a sigmoid or tanh to adjust the update (Zheng et al., 2024). AngularGrad scales steps based on the angle between two consecutive gradients to mitigate zig–zagging in the optimization trajectory (Roy et al., 2021). In contrast, our approach changes the backpropagation rule to approximate a parameter-path averaged gradient.

**Predicting future gradients.** Another approach that modifies the backpropagation phase in first–order optimization is *weight prediction* (Guan et al., 2024), which computes gradients at predicted future weights rather than at the current parameters. In this method, weights are first extrapolated forward according to several successive parameter updates, and a single backward pass is then performed at these predicted weights to obtain the update.

**Sharpness-aware optimization.** Many of the above methods can improve generalization by implicitly biasing optimization toward flatter local minima. The sharpness of a minimum is correlated with a model's generalization ability (Foret et al., 2021; Kim et al., 2022), and flatter minima tend to yield greater robustness to noise in the data domain. Sharpness-Aware Minimization (SAM) explicitly exploits this by adding an adversarial perturbation step and requires two backpropagations per iteration (Foret et al., 2021). SAM variants report significant generalization improvements (Foret et al., 2021; Kim et al., 2022).

**Integrated gradients in feature space.** Integrated Gradients (IG) applied to feature maps are widely used for neural network explainability (Sundararajan et al., 2017; Khorram et al., 2021; Sattarzadeh et al., 2021). These methods approximate the integral of the gradient along an input or feature-space path using a discrete Riemann sum (Hughes-Hallett et al., 2021), which is computationally inefficient for execution at every training step. To the best of our knowledge, IG has not been used to train neural networks.

**Second–order optimizers.** Second–order optimizers aim to capture finer-grained curvature information than first–order methods. Top results have been obtained by SOAP (Vyas et al., 2024) and the lightweight Eva method (Zhang et al., 2023), in the line of work that includes Shampoo (Gupta et al., 2018) and K–FAC (Martens & Grosse, 2015a; Grosse & Martens, 2016). See Appendix I for more details.

**Contrast with our approach.** In contrast to our method, none of the cited algorithms modifies the backpropagation procedure itself or approximates an averaged gradient in terms of model parameters over a broader continuous range between two parameter states on a fixed training batch.

## 3 METHODS

### 3.1 COMPUTING MULTIPLE PARAMETER UPDATES AT ONCE

Let us consider $n$ successive weight updates on a fixed training batch, excluding momentum. For many optimizers with constant learning rate $\eta$,

$$
\begin{aligned}
\Delta\theta^{(t-n+1:t)} = \sum_{i=t-n+1}^{t} \Delta\theta^{(i)} = \sum_{i=t-n+1}^{t} -\eta\,\boldsymbol{H}^{(i)}\nabla_\theta\ell\big(\theta^{(i)}\big) &\approx -\eta\,\boldsymbol{H}^{(t-n+1)} \sum_{i=t-n+1}^{t} \nabla_\theta\ell\big(\theta^{(i)}\big) \\
&\approx -n\,\eta\,\boldsymbol{H}^{(t-n+1)} \int_0^1 \nabla_\theta\ell\big(\theta^{(t-n+1)} + x(\theta^{(t)} - \theta^{(t-n+1)})\big)\,dx, \\
&= -n\eta\,\boldsymbol{H}^{(t-n+1)} \underset{\vartheta\in[\theta^{(t-n+1)},\theta^{(t)}]}{\mathcal{AVG}} \nabla_\vartheta\ell(\vartheta)
\end{aligned}
$$

$$(1)$$

where $\boldsymbol{H}^{(i)}$ is a slowly varying preconditioner (an inherent component of optimizers such as RM-SProp (Tieleman & Hinton, 2012), Adam (Kingma & Ba, 2014), SOAP, Eva, K–FAC, and Shampoo; see Appendix E, which shows that these optimizers can be expressed as matrix multiplications). We refer to the integral as $\mathcal{AVG}_{(\cdot)}(\cdot)$ (Appendix A). Let $u$ be the time for one forward–backward pass; our approximation computes $\Delta\theta^{(t-n+1:t)}$ in $O(2u)$ time, versus $O(nu)$ for methods requiring $n$ backpropagations.

### 3.2 GRADIENT AVERAGING

In the further equations, we assume a fixed sampled batch, thus we do not write it explicitly as a parameter. In our algorithm, given a plain feedforward neural network, the average gradient is approximated and propagated according to (Anonymized, 2025):

$$
\begin{aligned}
\mathcal{AVG}_{\vartheta\in[\theta,\theta']} \nabla_{\vartheta_k}\ell(\vartheta) &\approx \mathcal{AVG}_{\vartheta\in[\theta,\theta']} \frac{\partial\boldsymbol{x}_k(\vartheta)}{\partial\vartheta_k}\, \mathcal{AVG}_{\vartheta\in[\theta,\theta']} \frac{\partial\boldsymbol{x}_{k+1}(\vartheta)}{\partial\boldsymbol{x}_k(\vartheta)}\cdot \\
&\mathcal{AVG}_{\vartheta\in[\theta,\theta']} \frac{\partial\boldsymbol{x}_{k+2}(\vartheta)}{\partial\boldsymbol{x}_{k+1}(\vartheta)} \cdot\ldots\cdot \mathcal{AVG}_{\vartheta\in[\theta,\theta']} \frac{\partial\boldsymbol{x}_n(\vartheta)}{\partial\boldsymbol{x}_{n-1}(\vartheta)}\, \mathcal{AVG}_{\vartheta\in[\theta,\theta']} \nabla_{\boldsymbol{x}_n(\vartheta)}\ell(\vartheta)
\end{aligned}
$$

$$(2)$$

where $\ell$ is the loss function, $\vartheta_k$ are the parameters of layer $k$, and $(\boldsymbol{x}_k, \boldsymbol{x}_{k+1}, \ldots, \boldsymbol{x}_n)$ are layer outputs, treating each activation as a distinct layer. The notation $\nabla_{\boldsymbol{x}} f$ denotes the gradient of $f$ with respect to $\boldsymbol{x}$, and $\frac{\partial\boldsymbol{f}}{\partial\boldsymbol{x}}$ denotes the Jacobian matrix of the vector-valued function $\boldsymbol{f}$, where the $i^{\text{th}}$ column corresponds to the gradient vector $\nabla_{\boldsymbol{x}} f_i$. The operator $\mathcal{AVG}$ is defined in Appendix A.

See Appendix B for the details about efficient approximation of each averaging term of Eq. 2. However, the calculations will be introduced also further in pseudocode.

The average gradients are propagated in the same manner as gradients in the standard backpropagation algorithm. Computation based on Eq. 2 is fast and memory-efficient because the procedure mirrors standard backpropagation, which is performed as:

$$
\nabla_{\theta_k}\ell = \frac{\partial\boldsymbol{x}_k}{\partial\theta_k} \cdot \frac{\partial\boldsymbol{x}_{k+1}}{\partial\boldsymbol{x}_k} \cdot\ldots\cdot \frac{\partial\boldsymbol{x}_n}{\partial\boldsymbol{x}_{n-1}} \cdot \nabla_{\boldsymbol{x}_n}\ell,
$$

$$(3)$$

where "·" denotes standard matrix multiplication.

We use multiple algorithm versions for different purposes: (a) **AG–1:** the averaging range is between the parameter state before an update and the state after a single update forecasted by the base optimizer (Alg. 2, Appendix C). The sign of the parameter update matches the sign of the average gradient. This variant was introduced in (Anonymized, 2025). We use it to assess whether the averaged gradient efficiently minimizes the loss for a given batch. Since the averaging range is not spanned by the final applied update, this evaluates how universal the averaged-gradient direction is for batch loss reduction. (b) **AG–2:** identical to AG–1, but the step uses the averaged gradient (magnitude and direction), rather than only its sign (Alg. 2, Appendix C). Other properties remain

consistent with AG–1. (c) **Linear variants:** methods with "linear" in the name use Eq. 11 (in Appendix B) instead of Eq. 10, to approximate the averaged gradient through linear layers (fully connected, convolutional) more accurately. (d) **AG–3:** the averaging range spans multiple past updates (Alg. 1). Each new step refines several previous updates using broader loss–landscape information, in contrast to standard gradient descent which optimizes only at the current parameters without explicitly accounting for earlier updates. (e) **AG–3–1:** the AG–3 variant in which the averaging range is spanned by the most recent weight update.

The source code for this project is publicly available on GitHub at [ANONYMIZED for ICLR2026 review].

---

**Algorithm 1** *Single step in the AG–3 variant* for a plain feedforward neural network.

---

**Input:** Model with $n$ layers $f = (f_1, f_2, \ldots, f_n)$, Parameters of the model $\theta^{(t)}$, $optimizer$'s state $s^{(t)}$,
    Number of iterations corresponding to the minimal width of the averaging range $d$

**Output:** $\theta^{(t+1)}, s^{(t+1)}$

1: Sample a batch $B^{(t)}$.
2: $(\boldsymbol{x}_1^{(t)}, \boldsymbol{x}_2^{(t)}, \ldots, \boldsymbol{x}_n^{(t)}) = f(\theta^{(\lfloor t/d \rfloor \cdot d - d)}, B^{(t)})$ {Compute outputs of each layer of the model. Assume that $\boldsymbol{x}_i$ is a vector. Use the saved weights from iteration number $(\lfloor t/d \rfloor \cdot d - d)$ to retrieve parameters from $d$ to $2d$ iterations prior. If $(\lfloor t/d \rfloor \cdot d - d) < 0$, then use $\theta^{(0)}$.}
3: $(\boldsymbol{x}_1'^{(t)}, \boldsymbol{x}_2'^{(t)}, \ldots, \boldsymbol{x}_n'^{(t)}) = f(\theta^{(t)}, B^{(t)})$ {Compute the same for $\theta^{(t)}$.}
4: $G_{\boldsymbol{x},n}'^{(t)} \leftarrow \nabla_{\boldsymbol{x}_n'} \ell(\boldsymbol{x}_n', B^{(t)})$ {Compute the gradient of the last layer using new weights}
5: **for** $i \leftarrow n$ **to** 1 **step** $-1$ **do** {Iterate over layers to compute estimated average gradient.}
6:    **if** $f_i$ is an activation **then** {$G_{\theta,i}'^{(t)} = \emptyset$}
7:       $G_{\boldsymbol{x},i-1}'^{(t)} \leftarrow \dfrac{\boldsymbol{x}_i'^{(t)} - \boldsymbol{x}_i^{(t)}}{\boldsymbol{x}_{i-1}'^{(t)} - \boldsymbol{x}_{i-1}^{(t)}} \circ G_{\boldsymbol{x},i}'^{(t)}$ {Local approximation of the average gradient
        (Equation 8). If $(x_{i-1,j}'^{(t)} = x_{i-1,j}^{(t)})$, then $G_{\boldsymbol{x},(i-1,j)}'^{(t)} \leftarrow \dfrac{\partial x_{i,j}'^{(t)}}{\partial x_{i-1,j}'^{(t)}} G_{\boldsymbol{x},(i,j)}'^{(t)}$}
8:    **else** {Propagate gradient backwards without averaging}
9:       $G_{\boldsymbol{x},i-1}'^{(t)} \leftarrow \dfrac{\partial \boldsymbol{x}_i^{(t)}}{\partial \boldsymbol{x}_{i-1}^{(t)}} G_{\boldsymbol{x},i}'^{(t)}, \quad G_{\theta,i}'^{(t)} \leftarrow \dfrac{\partial \boldsymbol{x}_i^{(t)}}{\partial \theta_i^{(t)}} G_{\boldsymbol{x},i}'^{(t)}$ {Equation 10}
10:    **end if**
11: **end for**
12: $(\theta^{(t+1)}, s^{(t+1)}) \leftarrow optimizer(s^{(t)}, \theta^{(t)}, G_\theta'^{(t)})$ {Update weights and $optimizer$'s state}
13: **if** $t\%d = 0$ **then** {Once every $d$ iterations}
14:    Save $\theta^{(t)}$ to disk and remove $\theta^{(t-2d)}$. If $t \geq d$, load $\theta^{(t-d)}$ into host memory (CPU RAM) or device memory (GPU VRAM) depending on the execution target.
15: **end if**

---

See Appendix F for a comparison of computational time and memory usage across the evaluated methods.

## 4 EXPERIMENTS

### 4.1 EXPERIMENTAL SETUP

We test our method on multiple model architectures based on convolutional and fully–connected layers with different activation functions (see Tab. 1). The reasons for choosing these architectures are: (1) **Model A**: Shallow CNN used to assess learning in a loss landscape with less pronounced nonlinearity than deeper counterparts. (2) **Model B**: Tractable optimization task engineered to exhibit Shattered Gradients (Balduzzi et al., 2017) and singularity–related effects (Oyedotun et al., 2021; Yasrab, 2019), via repeated blocks of 10 tanh units in the linear stack. (3) **Model C**: Computationally tractable NLP optimization task exhibiting singularity–related phenomena (Oyedotun et al., 2021; Yasrab, 2019). (4) **ResNet–152**: Selected due to: (a) *Implementation practicality*: Significantly easier than Vision Transformers (ViTs) (Dosovitskiy et al., 2021) to implement from

scratch within our framework for modifying backpropagation logic (see Section 6). (b) *Depth sensitivity*: High layer count complicates optimization (Oyedotun et al., 2021), motivating methods that scale robustly with depth. (c) *Baseline strength*: ResNets often outperform ViTs (Dosovitskiy et al., 2021) when training from scratch or without large–scale pretraining; in such settings, ViTs "yield modest accuracies of a few percentage points below ResNets of comparable size" (Dosovitskiy et al., 2021; Chen et al., 2022b). (d) *Headroom for generalization*: SAM typically yields smaller generalization gains on ResNets than on ViTs (Chen et al., 2022a), underscoring the need for complementary methods. (e) *Maturity*: Well–established and well–understood architecture.

Table 1: *All model architectures investigated in this study.*

| Model | Architecture Summary | Datasets | Parameter Count |
|---|---|---|---|
| Model A | 6×Convolution 2D + ELU
1×Linear | MNIST,
Fashion MNIST | 17 506 |
| Model B | 2×Convolution 2D + ELU
2×Max Pooling 2D
28×Linear + Tanh | MNIST,
Fashion MNIST | 8 228 |
| Model C | 41×Convolution 2D + Tanh
5×Linear + Tanh | IMDb
(Maas et al., 2011) | 14 397 |
| ResNet–152
ResNet–152–GELU
ResNet–152–SiLU
ResNet–152–Sig | 151×Convolution 2D + ReLU/GELU/
SiLU/Sigmoid (with Batch Normalization)
1×Max Pooling 2D, 1×Avg Pooling 2D
1×Linear | Imagenet–OOD
(Yang et al., 2023)
Fine–Tuning | 60 192 808 |

We implemented our method in multiple optimizers: (a) SOAP, a promising second–order method that maintains a higher–dimensional empirical Fisher information matrix (eFIM; see Appendix I) approximation than Eva, at higher computational cost (Zhang et al., 2023; Vyas et al., 2024). Although SOAP was evaluated primarily on LLM training in the original paper (Vyas et al., 2024), we observed strong generalization across most of our experiments. (b) Adam. (c) RMSProp.

We conducted extensive learning–rate searches for baseline methods and selected variants, targeting average training–loss minimization to assess sample efficiency. Importantly, we report *all* tested models rather than cherry–picking best–performing ones. For ResNet–152, we additionally tuned SOAP using test–loss minimization to verify consistency with the training–loss search. Unless stated otherwise, our algorithms use learning rates *optimal for the gradient direction*, biasing comparisons in favor of gradient baselines. Final trainings were repeated up to 200 times per (model, dataset, optimizer) combination.

The main metrics we compute are: (a) **Relative sample efficiency:** For two algorithms, divide the best-loss epoch of one by the matching-loss epoch of the other. Compute this ratio for both mean and median losses, and report the geometric mean of the two ratios. Repeat over all contiguous epoch ranges that start at epoch 1 and report the arithmetic average of the resulting ratios. (b) **Relative batch-loss minimization improvement:** For each training step, compare batch-loss improvement across two optimizers at the same parameters on a fixed batch, by constraining the L1 norm of the parameter update vector ($\|\Delta\theta\|_1$) to be equal for both. (c) **Accuracy.**

Experiments ran on NVIDIA RTX 3070 8GB, NVIDIA A100 80GB, and NVIDIA H100 96GB GPUs; 80GB VRAM is sufficient for all experiments. See Appendix J for further details on experimental settings.

All error bars and $\pm$ values indicate standard errors of the mean (SEM), computed across independent runs for each configuration.

### 4.2 RESULTS

Variants utilizing forward–looking updates, AG–1 and AG–2, achieved strong *batch*-loss minimization improvements compared to all tested gradient–based optimizers across all tested deep models (see the "Relative Batch–Loss Minimization Improvement" column in Tabs. 2, 3, and 4). After switching ReLU activations to sigmoid on ResNet–152, these improvements were preserved for AG–1 and AG–2 (Tab. 5, Appendix D). Across all deep models, improvements range from $(2.23\pm0.01)\%$

to $(38.5 \pm 4.7)\%$, measured at matched update step lengths, as indicated by the "Relative Avg. Step Length" column in Tabs. 2, 3, 4, and 5.

Table 2: *Comparison of training efficiencies of various algorithms on Model B* (fully connected deep neural network with Tanh activations). $\pm$ values indicate SEM across independent runs.

| Dataset | Training Algorithm | Relative[1] Batch–Loss Minimization Improvement | Relative[1] Sample Efficiency and Baseline Algorithm | LR Optimized For | Relative[1] Avg. Step Length L1 and L2 | |
|---|---|---|---|---|---|---|
| MNIST | RMSProp AG–1 | $(11.3 \pm 1.8)\%$ | **444**% (of RMSProp) **393**% (of Adam) | AG–1 | 100% | 100% |
| | RMSProp AG–1 Linear | $(38.5 \pm 4.7)\%$ | 67.2% (of RMSProp) | AG–1 | 100% | 100% |
| | RMSProp AG–2 | $(30.0 \pm 2.8)\%$ | **399**% (of RMSProp) | AG–1 | 100% | 99.7% |
| | Soap AG–2 | $(11.9 \pm 0.2)\%$ | **132**% (of Soap) **1213**% (of Adam) | Gradient | 100% | 98.9% |
| | Soap AG–2 Linear | $(53.5 \pm 0.1)\%$ | 24.9% (of Soap) | Gradient | 100% | 98.0% |
| Fashion MNIST | RMSProp AG–1 | $(15.0 \pm 1.7)\%$ | **474**% (of RMSProp) **436**% (of Adam) | AG–1 | 100% | 100% |
| | RMSProp AG–1 Linear | $(18.0 \pm 3.1)\%$ | 0% (of RMSProp) | AG–1 | 100% | 100% |
| | RMSProp AG–2 | $(23.2 \pm 2.2)\%$ | **439**% (of RMSProp) | AG–1 | 100% | 99.7% |
| | Soap AG–2 | $(4.76 \pm 0.08)\%$ | 83.7% (of Soap) | Gradient | 100% | 99.0% |
| | Soap AG–2 Linear | $(26.4 \pm 0.8)\%$ | 32.8% (of Soap) | Gradient | 100% | 98.6% |

Table 3: *Efficiency comparison of training algorithms for Model C* (a deep convolutional neural network with Tanh activations trained on the IMDb dataset). $\pm$ denotes SEM across runs.

| Training Algorithm | Relative[1] Batch–Loss Minimization Improvement | Relative[1] Sample Efficiency and Baseline Algorithm | LR Optimized For | Relative[1] Avg. Step Length L1 and L2 | |
|---|---|---|---|---|---|
| RMSProp AG–1 | $(3.94 \pm 0.53)\%$ | **135**% (of RMSProp) **109**% (of Adam) | AG–1 | 100% | 100% |
| RMSProp AG–1 Linear | $(7.34 \pm 1.99)\%$ | 71.6% (of RMSProp) | AG–1 | 100% | 100% |
| RMSProp AG–2 | $(3.99 \pm 0.62)\%$ | 81.5% (of RMSProp) | AG–1 | 100% | 100% |
| Soap AG–2 | $(4.47 \pm 1.87)\%$ | 31.7% (of Soap) | Gradient | 100% | 101% |
| Soap AG–2 Linear | $(32.9 \pm 6.1)\%$ | 11.2% (of Soap) | Gradient | 100% | 98.6% |

Table 4: *Efficiency comparison of training algorithms for ResNet–152.* $\pm$ denotes SEM across runs.

| Training Algorithm | Relative[1] Batch–Loss Minimization Improvement | Relative[1] Sample Efficiency and Baseline Algorithm | LR Optimized For | Relative[1] Avg. Step Length L1 and L2 | |
|---|---|---|---|---|---|
| RMSProp AG–1 | $(2.23 \pm 0.01)\%$ | 69.9% (of RMSProp) | Gradient | 100% | 100% |
| RMSProp AG–1 Linear | $(7.07 \pm 0.08)\%$ | 0% (of RMSProp) | Gradient | 100% | 100% |
| RMSProp AG–2 | $(3.92 \pm 0.02)\%$ | 0% (of RMSProp) | Gradient | 100% | 99.8% |
| RMSProp AG–2 Linear | $(8.06 \pm 1.30)\%$ | 0% (of RMSProp) | Gradient | 100% | 99.4% |
| Adam AG–2 | $(3.02 \pm 0.01)\%$ | 99.7% (of Adam) | Gradient | 100% | 100% |
| Adam AG–2 Linear | $(3.50 \pm 0.01)\%$ | 98.1% (of Adam) | Gradient | 100% | 100% |
| Soap AG–1 | $(11.4 \pm 0.0)\%$ | 27.9% (of Soap) | Gradient | 100% | 100% |
| Soap AG–1 Linear | $(6.16 \pm 0.14)\%$ | 0% (of Soap) | Gradient | 100% | 100% |
| Soap AG–2 | $(5.32 \pm 0.01)\%$ | 36.6% (of Soap) | Gradient | 100% | 100% |
| Soap AG–2 Linear | $(13.8 \pm 0.62)\%$ | 0% (of Soap) | Gradient | 100% | 102% |

---

[1]All relative metrics are reported relative to gradient–based counterpart methods.

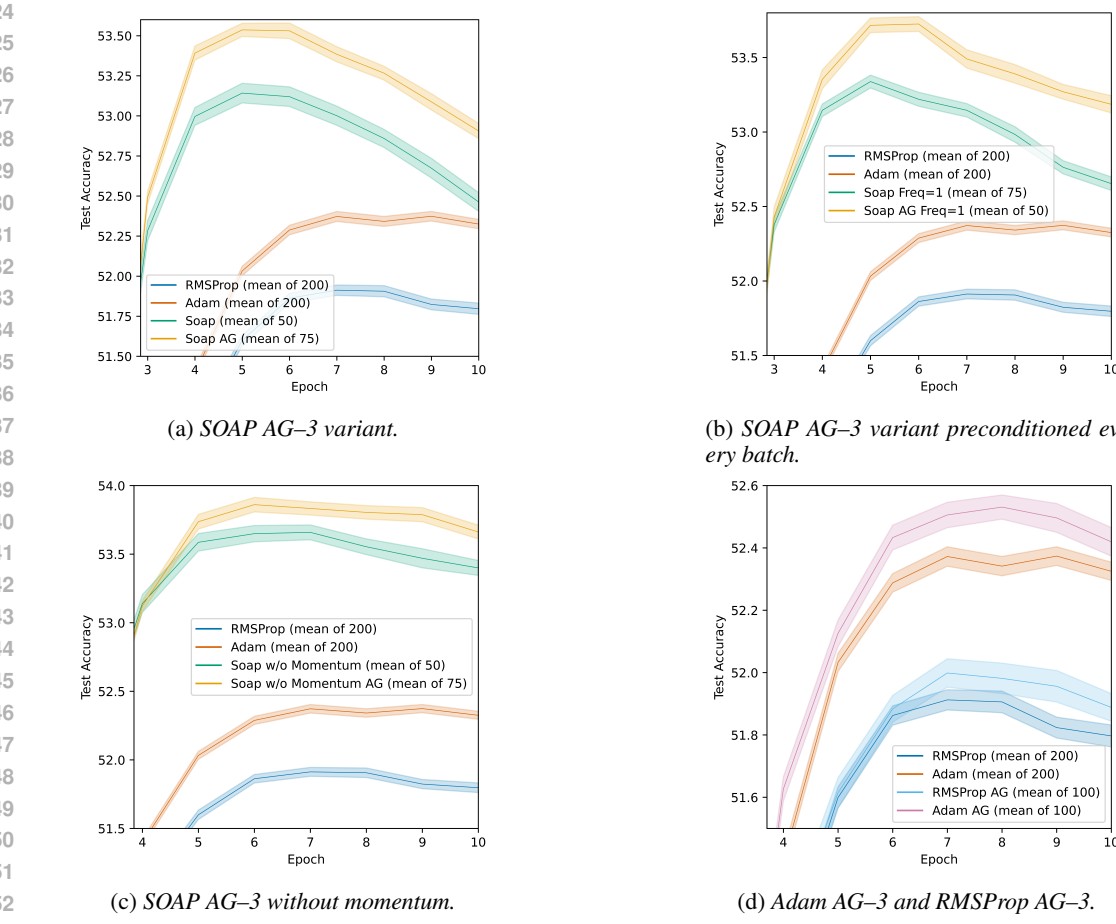

(a) *SOAP AG–3 variant.*

(b) *SOAP AG–3 variant preconditioned every batch.*

(c) *SOAP AG–3 without momentum.*

(d) *Adam AG–3 and RMSProp AG–3.*

Figure 1: *Test accuracy plots of ResNet–152* with confidence ranges (SEM). Each plot includes Adam and RMSProp as additional baseline algorithms. In all presented experiments, learning rates are optimized for gradient-based optimizers.

For shallow Model A on MNIST and Fashion MNIST, the AG–1, AG–2, AG–3–1, and AG–3 variants using RMSProp or SOAP achieved performance comparable to gradient-based counterparts across all measured metrics.

Sample efficiency indicates the speed of loss minimization over the full training dataset. Forward–looking updates (AG–1, AG–2) yielded mixed outcomes across experiments. High sample–efficiency gains were observed for several RMSProp runs on Models B and C (Tabs. 2, 3). For example, on Model B with RMSProp, AG–1 improved sample efficiency by more than $4.5\times$ on average, exceeding Adam by over $4\times$ (Tab. 2; Figs. 2a, 2c), at approximately $2\times$ the per–epoch compute relative to gradient–based RMSProp (see Appendix F for computational and memory comparisons). For Model B on MNIST, SOAP's sample efficiency improved by 32% (Tab. 2; Fig. 2a), with a corresponding reduction in test loss (Fig. 2b). In other settings, SOAP with AG–2 underperformed its baseline. Despite this variability, the forward–looking variants consistently meet their mathematical objective: efficient batch–loss minimization at matched step lengths.

Backward–looking updates (AG–3) tend to improve generalization on ResNet–152 (ReLU) across all tested optimizers and experimental setups (see Fig. 1). Accuracy gains of about $0.4\%$ were achieved for SOAP AG–3 (Fig. 1a) and for AG–3 paired with SOAP preconditioning applied every batch (Fig. 1b). Although the improvement may appear modest, it is nearly equal to the gain provided by momentum in Adam (Fig. 1d); note that Adam without adaptive momentum corresponds to RMSProp (Kingma & Ba, 2014; Tieleman & Hinton, 2012). Moreover, the improvement is roughly one third of the accuracy gap between first–order and second–order optimization observed in our experiments, despite using the optimal learning rate for the gradient.

Higher generalization improvements on ResNet–152 occur when momentum is paired with our gradient averaging, and they are also more pronounced with SOAP. These experiments suggest that the approximation of averaged gradients preserves acceptable accuracy across a wide range of 300–600 past updates. In the respective experiments, we also observed modest sample–efficiency improvements ranging from $0.6\%$ to $2.9\%$, all statistically significant.

Despite our averaged–gradient implementation not being computationally optimized, SOAP with AG–3 preconditioned at every batch is approximately $21\%$ slower than its gradient–based counterpart on ResNet–152 (see Appendix F for additional comparisons). Notably, this configuration has been reported as a strong generalization setting for SOAP on certain large language models (Vyas et al., 2024).

Using standard gradients, we fine–tuned ResNet–152 with Adam and RMSProp (each with separate learning–rate searches) under SiLU and GELU activations, both of which generally preserve non–zero gradients across their domains. In these models, the test accuracy of gradient–based Adam matched that of the ReLU–based ResNet–152. By contrast, replacing ReLU with either SiLU or GELU increased the test accuracy of gradient–based RMSProp, although the results did not surpass Adam across any of the three activation functions when using standard gradients. This suggests that preserving non–zero gradients may not provide additional benefit when training with the strongest first–order optimizer, Adam. Nevertheless, these experiments are not fully conclusive, since ResNet–152 was pretrained with ReLU, which may bias fine–tuning performance in favor of ReLU activations.

For other models, AG–3 maintained stable learning and did not visibly worsen performance relative to gradient baselines, except when the averaging range was excessively wide. AG–3–1 improved sample efficiency in selected scenarios—for example, with RMSProp on Model B (see Fig. 2, Appendix G).

Across all experiments, despite the "Linear" variants achieving more efficient batch–loss minimization, they yielded worse overall performance on the full dataset (see Tabs. 2, 3, 4, 5).

## 5 DISCUSSION

Across all deep models tested, our forward–looking AG–1 and AG–2 variants consistently delivered significant *batch*-loss improvements over gradient-based baselines with both first– and second–order optimizers. These results indicate that the averaged gradient encodes useful local structure of the loss landscape that remains informative even when the update step is modified (as in our algorithms). Within short–horizon objectives, where the goal is to reduce loss in one or a few steps, our method can outperform Adam and SOAP under matched step lengths with learning rates tuned for gradient baselines.

This short–term advantage has practical implications. After reaching full context length in an in–context learning task (Wies et al., 2023) without solving the prompt, a single or a few training updates can be applied to correct errors accumulated across the context, improving immediate accuracy without full retraining. Similarly, efficient batch–loss reduction relative to step length makes the approach suitable for test–time adaptation to new tasks (Finn et al., 2017; Nichol et al., 2018; Raghu et al., 2020), where limited updates are available and rapid improvement is critical.

Backward–looking gradient averaging (AG–3) yields accuracy improvements on ResNet–152. These gains are noteworthy given that, across our ResNet–152 trainings, total accuracy increased by 35 absolute percentage points (pp), primarily via knowledge transfer (fine–tuning). Thus, a $0.4\,\text{pp}$ gain during *second–order* fine–tuning may be broadly comparable in magnitude under differing settings to the $1.2$–$1.9\,\text{pp}$ improvements reported for SAM on ResNet–152 under *first–order* ImageNet training (Foret et al., 2021). Moreover, fine–tuning tasks typically exhibit smaller error reductions with SAM than full ImageNet training (Foret et al., 2021), and improvements over second–order baselines are generally harder to obtain.

Relative to SAM, AG–3 is less computationally demanding because it avoids the per–step perturbation overhead (i.e., the extra backward pass needed to compute weight perturbation); see Appendix F. Furthermore, AG–3 admits efficient parallelization: the averaging step (second line of Alg. 2) can be computed on upcoming batches in parallel with the main training loop, leaving the

remaining procedure computationally equivalent to a typical update. The main trade–off is memory: maintaining an averaged gradient over past updates increases the memory footprint, which can be reduced by accumulating the averaged gradient iteratively over sub–batches.

The averaged gradient over multiple past updates captures local sharpness along the recent optimization trajectory. AG–3 can be interpreted as minimizing the *trajectory-averaged* loss over a short horizon, which encourages flatter slopes and, consequently, can bias updates toward flatter local minima—an effect broadly linked to improved generalization (Foret et al., 2021).

On ResNet–152, the per–batch preconditioned SOAP AG–3 (non–optimized) variant is approximately 21% slower than its gradient–based counterpart, and has been reported as a strong generalization setting for SOAP on some large language models (Vyas et al., 2024). In high–dimensional regimes or with computationally intensive second–order preconditioning, this additional averaging overhead remains modest in practice.

There are two interesting avenues for future work related to SAM: (a) Compute the average gradient over the range of SAM perturbations, thereby estimating the "true" average sharpness of the surrounding loss–landscape geometry. Importantly, incorporating the average gradient into SAM *does not increase computational cost* relative to SAM itself, though it does increase memory requirements (see Appendix F). (b) Compute the average gradient over the range between two perturbed parameter values originating from different training iterations, which could provide a trajectory–aware measure of sharpness and potentially stabilize optimization across steps.

Both forward–looking (AG–1, AG–2) and backward–looking (AG–3) updates achieved more than a $4\times$ reduction in the number of epochs on two datasets for Model B under first–order optimization. These results support Eq. 1, indicating that multiple updates can be approximated by a single step using an averaged gradient estimated via a procedure based either on two backpropagations (Alg. 1) or a single backpropagation (Alg. 2). For Model B, the optimally implemented AG–3–1 in first–order experiments is estimated to require roughly $3\times$ less wall–clock time to convergence, with a modest memory overhead (see Appendix F) that can be mitigated via online gradient accumulation. Model B proved the most difficult to optimize among our tested architectures due to attenuated singularity issues (Oyedotun et al., 2021). Notably, we also observe significant sample–efficiency gains in first–order optimization for Model C.

Variants of our method that focus solely on gradient averaging at activations achieved better full–dataset performance across all experiments compared to the "Linear" variants. A plausible mechanism is that averaging gradients across multiple parameter vectors (equivalently, taking the elementwise average of per–layer diagonal activation Jacobians) can turn diagonal entries that are zero at a given vector into positive values in the average, thereby increasing algebraic rank (each newly nonzero diagonal entry in a diagonal matrix adds one to the rank) and mitigating singularity problems (Oyedotun et al., 2021) such as rank deficiency and severe ill–conditioning. At the activation layer, elementwise averaging in the backward pass preserves every coordinate retained by any individual diagonal Jacobian. Equivalently, its kernel is the intersection of the individual kernels, so more coordinate directions survive. Consequently, the chained Jacobian factors in Eq. 3 transmit signal along more directions unless later layers annihilate those coordinates. This effect depends on the involved parameter vectors inducing complementary activation patterns and on the absence of near–exact cancellation; the latter holds automatically at the diagonal stage for activations with nonnegative derivatives (e.g., ReLU, sigmoid, tanh).

For the shallow Model A, the gradient is approximately equal to the average gradient, thereby accounting for the comparable performance observed across methods.

# 6 LIMITATIONS

A primary limitation of our work is the implementation complexity required to ensure computational efficiency and portability across diverse architectures. We did not reimplement attention layers (Vaswani et al., 2017) within our framework or extend to full Transformer models, owing to the need to reconstruct the gradient–computation graph for each architecture and the absence of tooling to automate that process. Such extensive reimplementations are time–consuming and may affect reproducibility due to potential algorithmic and model divergences, as well as implementation instability.

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

## A  DEFINITION OF AVERAGE GRADIENT/JACOBIAN

Let us define the average gradient of a function $f : \mathbb{R}^n \to \mathbb{R}$ over a parameter range $[\boldsymbol{a}, \boldsymbol{b}]$, with $\boldsymbol{a}, \boldsymbol{b} \in \mathbb{R}^m$, via the mapping $\boldsymbol{g} : \mathbb{R}^m \to \mathbb{R}^n$, by

$$\underset{\boldsymbol{x} \in [\boldsymbol{a},\boldsymbol{b}]}{\mathcal{AVG}} \nabla_{\boldsymbol{g}(\boldsymbol{x})} f = \int_0^1 \nabla_{\boldsymbol{g}(\boldsymbol{a}+t \cdot (\boldsymbol{b}-\boldsymbol{a}))} f \; dt \tag{4}$$

which can alternatively be written by switching the integration variable to any component $x_i$ of $\boldsymbol{x}$, via $x_i = a_i + t \cdot (b_i - a_i)$, yielding

$$\underset{\boldsymbol{x} \in [\boldsymbol{a},\boldsymbol{b}]}{\mathcal{AVG}} \nabla_{\boldsymbol{g}(\boldsymbol{x})} f = \frac{1}{b_i - a_i} \cdot \int_{a_i}^{b_i} \nabla_{\boldsymbol{g}(\boldsymbol{x})} f \; dx_i \tag{5}$$

However, any case in which a vector component would cause division by zero should be handled via Eq. 4.

If $\boldsymbol{f}(\boldsymbol{x}) : \mathbb{R}^n \to \mathbb{R}^l$, then by applying Eq. 4 component-wise we get

$$\underset{\boldsymbol{x} \in [\boldsymbol{a},\boldsymbol{b}]}{\mathcal{AVG}} \frac{\partial \boldsymbol{f}}{\partial \boldsymbol{g}(\boldsymbol{x})} = [\, \underset{\boldsymbol{x} \in [\boldsymbol{a},\boldsymbol{b}]}{\mathcal{AVG}} \nabla_{\boldsymbol{g}(\boldsymbol{x})} f_1, \underset{\boldsymbol{x} \in [\boldsymbol{a},\boldsymbol{b}]}{\mathcal{AVG}} \nabla_{\boldsymbol{g}(\boldsymbol{x})} f_2, \dots, \underset{\boldsymbol{x} \in [\boldsymbol{a},\boldsymbol{b}]}{\mathcal{AVG}} \nabla_{\boldsymbol{g}(\boldsymbol{x})} f_l]$$
$$= [\int_0^1 \nabla_{\boldsymbol{g}(\boldsymbol{a}+t \cdot (\boldsymbol{b}-\boldsymbol{a}))} f_1 \; dt, \int_0^1 \nabla_{\boldsymbol{g}(\boldsymbol{a}+t \cdot (\boldsymbol{b}-\boldsymbol{a}))} f_2 \; dt, \dots, \int_0^1 \nabla_{\boldsymbol{g}(\boldsymbol{a}+t \cdot (\boldsymbol{b}-\boldsymbol{a}))} f_l \; dt] \tag{6}$$
$$= \int_0^1 \frac{\partial \boldsymbol{f}}{\partial \boldsymbol{g}(\boldsymbol{a} + t \cdot (\boldsymbol{b}-\boldsymbol{a}))} \; dt$$

or, alternatively, using Eq. 5:

$$
\begin{aligned}
\mathcal{AVG}_{\boldsymbol{x}\in[\boldsymbol{a},\boldsymbol{b}]} \frac{\partial \boldsymbol{f}}{\partial \boldsymbol{g}(\boldsymbol{x})} &= [\, \mathcal{AVG}_{\boldsymbol{x}\in[\boldsymbol{a},\boldsymbol{b}]} \nabla_{\boldsymbol{g}(\boldsymbol{x})}f_1, \, \mathcal{AVG}_{\boldsymbol{x}\in[\boldsymbol{a},\boldsymbol{b}]} \nabla_{\boldsymbol{g}(\boldsymbol{x})}f_2, \ldots, \, \mathcal{AVG}_{\boldsymbol{x}\in[\boldsymbol{a},\boldsymbol{b}]} \nabla_{\boldsymbol{g}(\boldsymbol{x})}f_l] \\
&= \frac{1}{b_i - a_i} \cdot [\int_{a_i}^{b_i} \nabla_{\boldsymbol{g}(\boldsymbol{x})}f_1 \; dx_i, \int_{a_i}^{b_i} \nabla_{\boldsymbol{g}(\boldsymbol{x})}f_2 \; dx_i, \ldots, \int_{a_i}^{b_i} \nabla_{\boldsymbol{g}(\boldsymbol{x})}f_l \; dx_i] \\
&= \frac{1}{b_i - a_i} \int_{a_i}^{b_i} \frac{\partial \boldsymbol{f}}{\partial \boldsymbol{g}(\boldsymbol{x})} \; dx_i
\end{aligned}
\tag{7}
$$

which holds for every $i$. However, if $b_i = a_i$, Eq. 6 must be used instead.

## B  COMPUTING AVERAGE GRADIENT

Our aim is to compute Eq. 2 efficiently and, where possible, with high precision. The calculations up to Eq. 10 were introduced in our previous work (Anonymized, 2025).

Assume that the averaging interval is $[\theta, \theta']$; see Appendix A for the formal definition, which aligns with the intuitive notion of averaging across parameter values. The averaged derivatives of each scalar, nonlinear activation are computed as follows:

$$
\begin{aligned}
\mathcal{AVG}_{\vartheta\in[\theta,\theta']} \frac{\partial x_{k,i}(\vartheta)}{\partial x_{k-1,i}(\vartheta)} &= \int_0^1 \frac{\partial x_{k,i}(\theta + t\cdot(\theta'-\theta))}{\partial x_{k-1,i}(\theta + t\cdot(\theta'-\theta))} dt \\
&= \frac{\int_0^1 \frac{\partial x_{k,i}(\theta+t\cdot(\theta'-\theta))}{\partial x_{k-1,i}(\theta+t\cdot(\theta'-\theta))} dt \cdot \int_0^1 \frac{\partial x_{k-1,i}(\theta+t\cdot(\theta'-\theta))}{\partial t} dt}{\int_0^1 \frac{\partial x_{k-1,i}(\theta+t\cdot(\theta'-\theta))}{\partial t} dt}
\end{aligned}
$$

where $x_{k-1,i}$ and $x_{k,i}$ are the input and output of an activation, respectively. Both variables depend on the complete set of model parameter values, denoted by $\vartheta$.

We approximate $x_{k-1,i}(\theta + t\cdot(\theta' - \theta))$ as an affine (first-order) function of $t$, i.e. $x_{k-1,i}(\theta + t\cdot(\theta' - \theta)) \approx \alpha\, t + \beta$, which implies $\frac{\partial x_{k-1,i}}{\partial t} = \alpha = \text{const}$ and therefore $\int_0^1 \frac{\partial x_{k-1,i}}{\partial t}\, dt = \int_0^1 \alpha\, dt = \alpha = \frac{\partial x_{k-1,i}}{\partial t}$. Finally, this constant can be carried into the remaining integral $\int_0^1 \frac{\partial x_{k,i}}{\partial x_{k-1,i}}\, dt$, yielding:

$$
\begin{aligned}
\mathcal{AVG}_{\vartheta\in[\theta,\theta']} \frac{\partial x_{k,i}(\vartheta)}{\partial x_{k-1,i}(\vartheta)} &\approx \frac{\int_0^1 \frac{\partial x_{k,i}(\theta+t\cdot(\theta'-\theta))}{\partial x_{k-1,i}(\theta+t\cdot(\theta'-\theta))} \cdot \frac{\partial x_{k-1,i}(\theta+t\cdot(\theta'-\theta))}{\partial t} dt}{\int_0^1 \frac{\partial x_{k-1,i}(\theta+t\cdot(\theta'-\theta))}{\partial t} dt} \\
&= \frac{\int_0^1 \frac{\partial x_{k,i}(\theta+t\cdot(\theta'-\theta))}{\partial t} dt}{\int_0^1 \frac{\partial x_{k-1,i}(\theta+t\cdot(\theta'-\theta))}{\partial t} dt} = \frac{x_{k,i}(\theta') - x_{k,i}(\theta)}{x_{k-1,i}(\theta') - x_{k-1,i}(\theta)}
\end{aligned}
\tag{8}
$$

The main advantage of our approximation versus alternative schemes is that we never linearize the partial derivative $\frac{\partial x_{k,i}}{\partial x_{k-1,i}}$ in $\theta$, $\theta'$, or $t$; consequently, it captures the true dynamics of the gradients more accurately. Moreover, the calculation is computationally efficient. Division–by–zero cases are handled as described in Algorithms 1 and 2.

We assume that each activation is represented as a distinct $k^{\text{th}}$ layer, i.e., $\boldsymbol{f}_k : \mathbb{R}^m \to \mathbb{R}^m$, where $m$ is the length of both representations $\boldsymbol{x}_{k-1}$ and $\boldsymbol{x}_k$. Its input is the output from the $(k-1)$th layer, given by $\boldsymbol{x}_{k-1}(\vartheta) = [x_{k-1,1}(\vartheta), x_{k-1,2}(\vartheta), ..., x_{k-1,n}(\vartheta)]$ which depends on the model parameters $\vartheta$. Then the average gradient of layer $\boldsymbol{f}_k$ is given by:

$$
\begin{aligned}
\mathcal{AVG}_{\vartheta\in[\theta,\theta']} \frac{\partial \boldsymbol{x}_k(\vartheta)}{\partial \boldsymbol{x}_{k-1}(\vartheta)} = \text{diag}([&\mathcal{AVG}_{\vartheta\in[\theta,\theta']} \frac{\partial x_{k,1}(\vartheta)}{\partial x_{k-1,1}(\vartheta)}, \mathcal{AVG}_{\vartheta\in[\theta,\theta']} \frac{\partial x_{k,2}(\vartheta)}{\partial x_{k-1,2}(\vartheta)}, \ldots, \\
&\mathcal{AVG}_{\vartheta\in[\theta,\theta']} \frac{\partial x_{k,n}(\vartheta)}{\partial x_{k-1,n}(\vartheta)}])
\end{aligned}
\tag{9}
$$

where each term $\mathcal{AVG}_{(\cdot)}(\cdot)$ is approximated in Eq. 8, and $\text{diag}(\cdot)$ denotes a diagonal matrix constructed from the input vector.

For method variants without "linear" in their names, we do not alter Jacobian terms associated with affine or convolutional layers; instead, our modifications apply only to the backward pass through

activation functions. Assuming that the $k^{\text{th}}$ layer is a pointwise activation and that the $(k-1)$th layer is either fully connected or convolutional, we extend the approximation of Eq. 2 accordingly:

$$\mathcal{AVG}_{\vartheta \in [\theta,\theta']} \frac{\partial \boldsymbol{x}_k(\vartheta)}{\partial \boldsymbol{x}_{k-2}(\vartheta)} \approx \frac{\partial \boldsymbol{x}_{k-1}(\theta)}{\partial \boldsymbol{x}_{k-2}(\theta)} \mathcal{AVG}_{\vartheta \in [\theta,\theta']} \frac{\partial \boldsymbol{x}_k(\vartheta)}{\partial \boldsymbol{x}_{k-1}(\vartheta)}$$
$$\mathcal{AVG}_{\vartheta \in [\theta,\theta']} \frac{\partial \boldsymbol{x}_k(\vartheta)}{\partial \vartheta_{k-1}} \approx \frac{\partial \boldsymbol{x}_{k-1}(\theta)}{\partial \theta_{k-1}} \mathcal{AVG}_{\vartheta \in [\theta,\theta']} \frac{\partial \boldsymbol{x}_k(\vartheta)}{\partial \boldsymbol{x}_{k-1}(\vartheta)} \quad (10)$$

This equation intuitively moves the approximated values closer to the gradient. Our goal is to obtain an approximation of the average gradient that is more accurate than using the gradient directly as an estimate. Therefore, if the precision of the approximation remains between that of the gradient and our current estimate in Eq. 2, it satisfies our objective.

As an alternative to the unaveraged Jacobian terms in Eq. 10, $\frac{\partial \boldsymbol{x}_{k-1}}{\partial \boldsymbol{x}_{k-2}}$ and $\frac{\partial \boldsymbol{x}_{k-1}}{\partial \theta_{k-1}}$, the average gradient for affine-parameterized layers can be approximated as:

$$\mathcal{AVG}_{\vartheta \in [\theta,\theta']} \frac{\partial \boldsymbol{x}_{k-1}(\vartheta)}{\partial \boldsymbol{x}_{k-2}(\vartheta)} \approx \frac{1}{2} \left( \frac{\partial \boldsymbol{x}_{k-1}(\theta)}{\partial \boldsymbol{x}_{k-2}(\theta)} + \frac{\partial \boldsymbol{x}_{k-1}(\theta')}{\partial \boldsymbol{x}_{k-2}(\theta')} \right)$$
$$\mathcal{AVG}_{\vartheta \in [\theta,\theta']} \frac{\partial \boldsymbol{x}_{k-1}(\vartheta)}{\partial \vartheta_{k-1}} \approx \frac{1}{2} \left( \frac{\partial \boldsymbol{x}_{k-1}(\theta)}{\partial \theta_{k-1}} + \frac{\partial \boldsymbol{x}_{k-1}(\theta')}{\partial \theta'_{k-1}} \right) \quad (11)$$

which is used by the variants labeled "linear".

See Algorithms 2 and 1 for details on computing the average gradient.

## C    PSEUDOCODE FOR FORWARD–LOOKING ALGORITHM VARIANTS

**Algorithm 2** *Single step of the forward–looking variants of our algorithm* assuming a plain feedforward neural network. Different algorithm settings are marked in colors (AG–1 (Anonymized, 2025), AG–2, and Linear).

---

**Input:** Model with $n$ layers $f = (f_1, f_2, \ldots, f_n)$, Parameters of the model $\theta^{(t)}$, $optimizer$'s state $s^{(t)}$
**Output:** $\theta^{(t+1)}, s^{(t+1)}$

---

1: Sample a batch $B^{(t)}$.
2: $(\boldsymbol{x}_1^{(t)}, \boldsymbol{x}_2^{(t)}, \ldots, \boldsymbol{x}_n^{(t)}) = f(\theta^{(t)}, B^{(t)})$ {Compute outputs of each layer of the model. Assume that $\boldsymbol{x}_i$ is a vector.}
3: $(G_\theta^{(t)}, G_{\boldsymbol{x}}^{(t)}) = \nabla_{(\theta^{(t)}, \boldsymbol{x}^{(t)})} \ell(\boldsymbol{x}_n^{(t)}, B^{(t)})$ {Calculate gradients.}
4: $(\theta'^{(t)}, s^{(t+1)}) \leftarrow optimizer(s^{(t)}, \theta^{(t)}, G_\theta^{(t)})$ {Estimate the next update.}
5: $(\boldsymbol{x}_1'^{(t)}, \boldsymbol{x}_2'^{(t)}, \ldots, \boldsymbol{x}_n'^{(t)}) = f(\theta'^{(t)}, B^{(t)})$
6: $G_{\boldsymbol{x},n}'^{(t)} \leftarrow \nabla_{\boldsymbol{x}_n'} \ell(\boldsymbol{x}_n', B^{(t)})$ {Compute the gradient of the last layer using new weights}
7: **for** $i \leftarrow n$ **to** $1$ **step** $-1$ **do** {Iterate over layers to compute estimated average gradient.}
8:     **if** $f_i$ is an activation **then** {$G_{\theta,i}'^{(t)} = \emptyset$}
9:         $G_{\boldsymbol{x},i-1}'^{(t)} \leftarrow \dfrac{\boldsymbol{x}_i'^{(t)} - \boldsymbol{x}_i^{(t)}}{\boldsymbol{x}_{i-1}'^{(t)} - \boldsymbol{x}_{i-1}^{(t)}} \circ G_{\boldsymbol{x},i}'^{(t)}$ {Equation 8. If $(x_{i-1,j}'^{(t)} = x_{i-1,j}^{(t)})$, then

        $G_{\boldsymbol{x},(i-1,j)}'^{(t)} \leftarrow \dfrac{\partial x_{i,j}^{(t)}}{\partial x_{i-1,j}^{(t)}} G_{\boldsymbol{x},(i,j)}'^{(t)}$}
10:     **else** {$f_i$ is a parameterized layer}
11:         **if** Algorithm variant is "Linear" **then** {Calculate average gradient of a parameterized layer}
12:             $G_{\boldsymbol{x},i-1}'^{(t)} \leftarrow \dfrac{1}{2}\left(\dfrac{\partial \boldsymbol{x}_i^{(t)}}{\partial \boldsymbol{x}_{i-1}^{(t)}} + \dfrac{\partial \boldsymbol{x}_i'^{(t)}}{\partial \boldsymbol{x}_{i-1}'^{(t)}}\right) G_{\boldsymbol{x},i}'^{(t)}$ {Equation 11}
13:             $G_{\theta,i}'^{(t)} \leftarrow \dfrac{1}{2}\left(\dfrac{\partial \boldsymbol{x}_i^{(t)}}{\partial \theta_i^{(t)}} + \dfrac{\partial \boldsymbol{x}_i'^{(t)}}{\partial \theta_i'^{(t)}}\right) G_{\boldsymbol{x},i}'^{(t)}$ {Equation 11}
14:             **if** Algorithm variant is AG–2 Linear **then**
15:                 Scale $G_{\boldsymbol{x},i-1}'^{(t)}$ and $G_{\theta,i}'^{(t)}$ to match the gradient length ($\left\|\dfrac{\partial \ell}{\partial \boldsymbol{x}_{i-1}^{(t)}}\right\|_2$ and

                $\left\|\dfrac{\partial \ell}{\partial \theta_i^{(t)}}\right\|_2$, respectively).
16:             **end if**
17:         **else** {Propagate gradient backwards without averaging}
18:             $G_{\boldsymbol{x},i-1}'^{(t)} \leftarrow \dfrac{\partial \boldsymbol{x}_i^{(t)}}{\partial \boldsymbol{x}_{i-1}^{(t)}} G_{\boldsymbol{x},i}'^{(t)}$ ,    $G_{\theta,i}'^{(t)} \leftarrow \dfrac{\partial \boldsymbol{x}_i^{(t)}}{\partial \theta_i^{(t)}} G_{\boldsymbol{x},i}'^{(t)}$ {Equation 10}
19:         **end if**
20:     **end if**
21: **end for**
22: **if** Algorithm variant is AG–1 **then**
23:     $\theta^{(t+1)} \leftarrow \theta^{(t)} + |\theta'^{(t)} - \theta^{(t)}| \circ \text{sign}(G_\theta'^{(t)})$
24: **else** {Algorithm variant is AG–2}
25:     $\theta''^{(t)} \leftarrow optimizer(s^{(t)}, \theta^{(t)}, G_\theta'^{(t)})$ {Update weights, but without modifying $optimizer$'s state}
26:     Scale the update $\Delta\theta = \theta''^{(t)} - \theta^{(t)}$ to match the gradient update length $\left\|\theta'^{(t)} - \theta^{(t)}\right\|_1$
27:     $\theta^{(t+1)} \leftarrow \theta^{(t)} + \Delta\theta$
28: **end if**

---

## D  RESULTS FOR FORWARD-LOOKING ALGORITHM VARIANTS ON RESNET-152 USING SIGMOID ACTIVATIONS

Table 5: *Efficiency comparison of training algorithms on ResNet-152 with sigmoid activations instead of ReLU.* A single empirically tuned learning rate is held constant across all experiments. All relative metrics are reported relative to gradient–based counterpart methods. $\pm$ values indicate SEM across independent runs.

| Training Algorithm | Relative Batch–Loss Minimization Improvement | Relative Sample Efficiency and Baseline Algorithm | Relative Avg. Step Length L1 and L2 | |
|---|---|---|---|---|
| RMSProp AG–1 | $(\mathbf{2.68 \pm 0.07})\%$ | 58.7% (of RMSProp) | 100% | 100% |
| RMSProp AG–2 | $(\mathbf{22.4 \pm 1.1})\%$ | 27.9% (of RMSProp) | 100% | 100% |
| RMSProp AG–2 Linear | $(-65.3 \pm 0.7)\%$ | 5.82% (of RMSProp) | 100% | 99.7% |
| Soap AG–2 | $(\mathbf{7.53 \pm 0.91})\%$ | 28.1% (of RMSProp) | 100% | 100% |
| Soap AG–2 Linear | $(-4.58 \pm 1.46)\%$ | 10.7% (of RMSProp) | 100% | 100% |

## E  PRECONDITIONING MATRICES OF DIFFERENT OPTIMIZERS

In Eq. 1 we assume that model-parameter updates of typical optimizers (RMSProp (Tieleman & Hinton, 2012), Adam (Kingma & Ba, 2014), SOAP (Vyas et al., 2024), Eva (Zhang et al., 2023), K-FAC (Martens & Grosse, 2015a), and Shampoo (Gupta et al., 2018)) admit the unified form $\Delta\theta = -\mu \, \boldsymbol{H} \, \nabla_\theta \ell$, i.e., a preconditioner $\boldsymbol{H}$ exists:

- **Adam/RMSProp (diagonal inverse square root):** $\boldsymbol{H} = \mathrm{diag}\big((\mathbb{E}[(\nabla_\theta \ell) \circ (\nabla_\theta \ell)] + \epsilon)^{-1/2}\big)$, where the square root and division are elementwise, $\epsilon > 0$ stabilizes the scaling, and $\circ$ denotes Hadamard multiplication.

- **K–FAC (structured inverse):** $\boldsymbol{H} \approx (\hat{\boldsymbol{F}} + \lambda I)^{-1}$, where $\hat{\boldsymbol{F}}$ is a low–dimensional Kronecker–factored surrogate of the empirical Fisher $\hat{\boldsymbol{F}} = \mathbb{E}[(\nabla_\theta \ell)(\nabla_\theta \ell)^\top]$ and $\lambda > 0$ provides damping.

- **Eva (spectral/diagonal inverse square root):** $\boldsymbol{H} \approx (\hat{\boldsymbol{F}} + \epsilon I)^{-1}$, where $\hat{\boldsymbol{F}}$ is a structured (spectral or diagonal) second–moment/Fisher surrogate; Eva uses AdaGrad–style normalization rather than a full inverse.

- **Shampoo (low–dimensional Fisher surrogate, inverse square root):** Shampoo constructs a low–dimensional surrogate $\hat{\boldsymbol{F}}$ of $\boldsymbol{F}$ (e.g., via Kronecker–structured contractions of $\mathbb{E}[(\nabla_\theta \ell)(\nabla_\theta \ell)^\top]$ along parameter dimensions) and applies inverse–square-root preconditioning: $\boldsymbol{H} \approx (\sum_t \hat{\boldsymbol{F}}^{(t)} + \epsilon I)^{-1/2}$. The standard power is $-\frac{1}{2}$, yielding second-order normalization consistent with AdaGrad–style scaling.

- **SOAP (Adam in a structured eigenbasis):**

$$\boldsymbol{H} = \boldsymbol{Q} \, \mathrm{diag}\left(\sqrt{\mathbb{E}\left[(\boldsymbol{Q}^\top \nabla_\theta \ell) \circ (\boldsymbol{Q}^\top \nabla_\theta \ell)\right]} + \epsilon\right)^{-1} \boldsymbol{Q}^\top,$$

where $\boldsymbol{Q}$ holds eigenvectors of a slowly varying structured surrogate of $\boldsymbol{F}$; SOAP performs Adam–style second-moment normalization per eigen–direction.

## F  COMPUTATIONAL AND MEMORY EFFICIENCY COMPARISON

Table 6: *Notation for time and memory* (per training step, per batch).

| Symbol | Meaning |
| --- | --- |
| $T_{\text{fwd}}$ | Forward pass time (wall--clock). |
| $T_{\text{bwd}}$ | Backward pass (backpropagation) time (wall--clock). |
| $T_{\text{opt}}$ | Optimizer step time, including the amortized per--step cost of any infrequent optimizer procedures (e.g., preconditioner updates). |
| $T_{\text{scl}}$ | Time to compute a norm-based scaling factor and apply it to all parameters or all gradient entries (same size). This comprises a reduction to compute the chosen norm (e.g., $L_2$ or $L_\infty$) and a subsequent scaling pass over the tensor(s). |
| $T_{prt}$ | SAM perturbation time (norm reduction over gradients followed by a scaling pass). For commonly used model sizes on modern accelerators these low-intensity operations are bandwidth-bound, so $T_{prt}$ is of the same order as $T_{\text{opt}}$ for plain SGD; it is smaller for SGD with momentum and strictly smaller for stateful or preconditioned optimizers (e.g., Adam, Shampoo/K–FAC). |
| $M_\theta$ | Memory for model parameters only. |
| $M_x$ | Activation/graph memory for one batch (intermediate tensors retained for backprop), excluding parameters and optimizer state. Note that typically $M_x \gg M_\theta$; however, $M_x$ can be reduced when accumulating gradients over a fraction of a batch. |
| $M_{\text{opt}}$ | Memory for optimizer state (e.g., moments, statistics, preconditioners). |
| $M_{\text{act}}$ | Memory for stored activation inputs only under recomputation (checkpointing) policies; otherwise refers to inputs and outputs. In all cases, $M_{\text{act}} \leq M_x$. |

Table 7: *Computational and memory efficiency comparison* (per training step, per batch). See Table 6 for notation.

| Training algorithm | Computational time | Memory requirements |
| --- | --- | --- |
| SGD, RMSProp, Adam, SOAP, Eva, Shampoo, K–FAC | $T_{\text{fwd}} + T_{\text{bwd}} + T_{\text{opt}}$ | $M_\theta + M_x + M_{\text{opt}}$ |
| Sharpness–Aware Minimization | $2T_{\text{fwd}} + 2T_{\text{bwd}} + T_{\text{opt}} + T_{\text{prt}}$ | $2M_\theta + M_x + M_{\text{opt}}$ |
| AG–3 | $2T_{\text{fwd}} + T_{\text{bwd}}^\dagger + T_{\text{opt}}$ | $2M_\theta^\ddagger + M_x + M_{\text{act}} + M_{\text{opt}}$ |
| AG–1 | $2T_{\text{fwd}} + 2T_{\text{bwd}}^\dagger + 2T_{\text{opt}}$ | $2M_\theta + M_x + M_{\text{act}} + M_{\text{opt}}$ |
| AG–1 (Linear) | $2T_{\text{fwd}} + 3T_{\text{bwd}}^\S + 2T_{\text{opt}}$ | $2M_\theta + 2M_x + M_{\text{opt}}$ |
| AG–2 | $2T_{\text{fwd}} + 2T_{\text{bwd}}^\dagger + 2T_{\text{opt}} + 2T_{\text{scl}}$ | $2M_\theta + M_x + M_{\text{act}} + M_{\text{opt}}$ |
| AG–2 (Linear) | $2T_{\text{fwd}} + 3T_{\text{bwd}}^\S + 2T_{\text{opt}} + 2T_{\text{scl}}$ | $2M_\theta + 2M_x + M_{\text{opt}}$ |
| SAM incorporating AG–3 | Same as SAM when the averaging range in AG–3 matches the perturbation range in SAM. | |

[†] Averaged backpropagation in AG–3 is implemented to run in roughly the time of one backpropagation.
[‡] Excludes additional $M_\theta$ associated with disk usage.
[§] Averaged backpropagation in "Linear" variants is two times slower due to the utilization of Eq. 11.

# G    PLOTS OF MODEL B

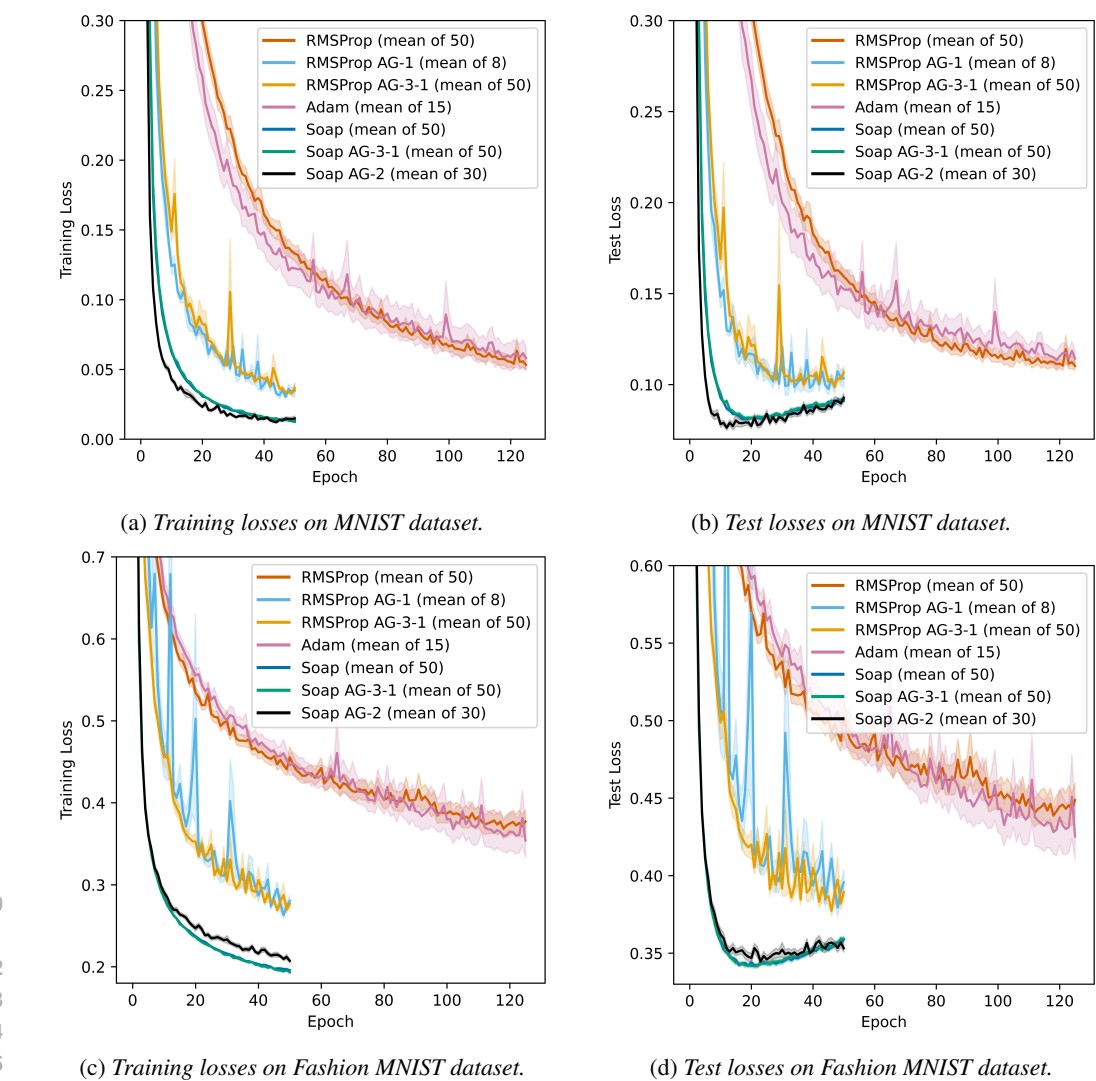

(a) *Training losses on MNIST dataset.*

(b) *Test losses on MNIST dataset.*

(c) *Training losses on Fashion MNIST dataset.*

(d) *Test losses on Fashion MNIST dataset.*

Figure 2: *Plots of Model B* with confidence ranges (SEM). The optimizers for which the learning rates are optimized are presented in Tab. 2.

## H   TRAINING AND TEST LOSS CURVES FOR MODEL C

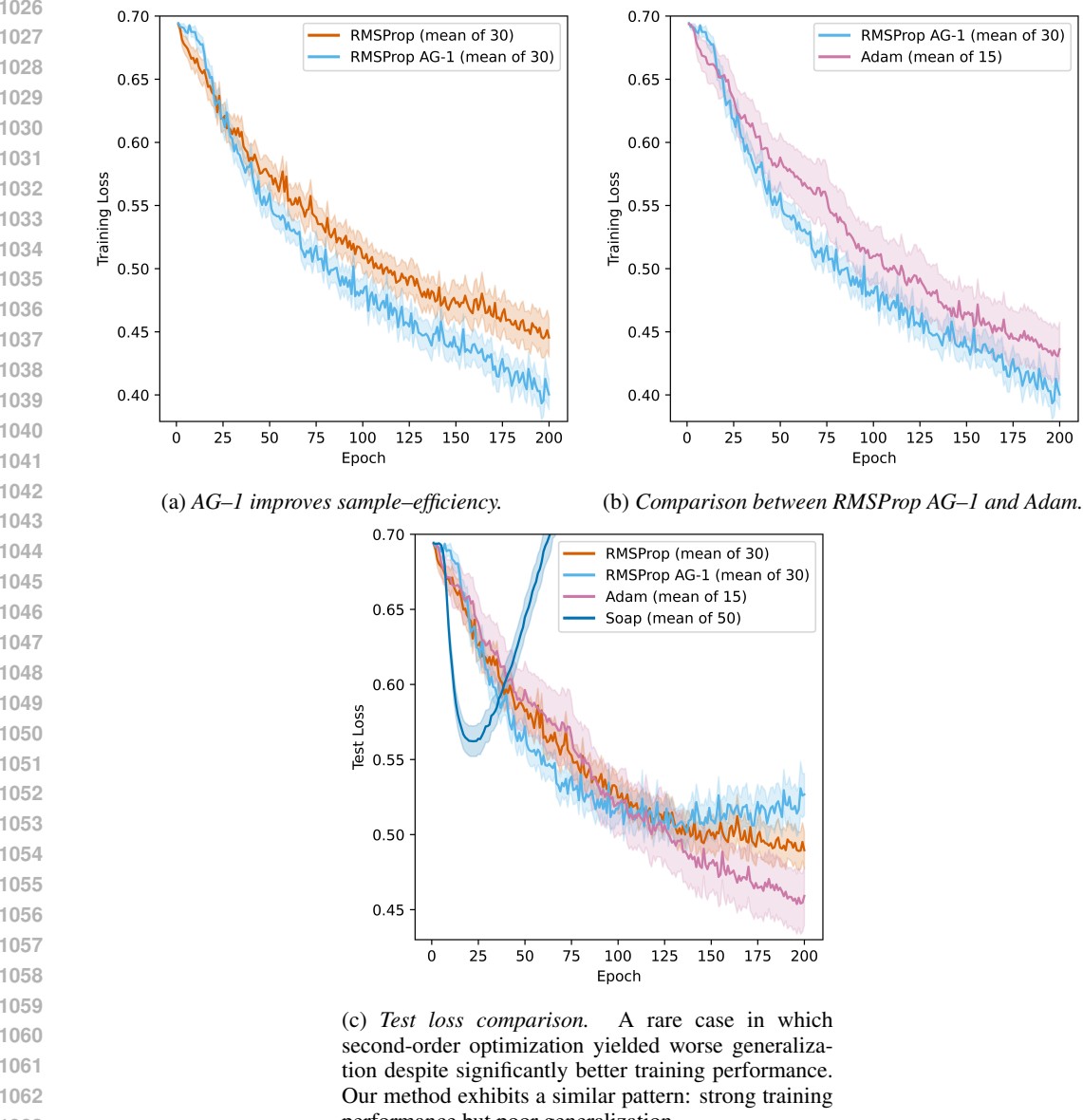

(a) *AG–1 improves sample–efficiency.*

(b) *Comparison between RMSProp AG–1 and Adam.*

(c) *Test loss comparison.* A rare case in which second-order optimization yielded worse generalization despite significantly better training performance. Our method exhibits a similar pattern: strong training performance but poor generalization.

Figure 3: *Model C plots with confidence ranges (SEM).* The objectives used for learning rate tuning are listed in Tab. 3.

# I   SECOND–ORDER OPTIMIZATION

The fundamental problem of second–order optimization is that computing all second–order derivatives requires $O(n^2)$ computational time for $n$ model parameters. Here, $O(\cdot)$ denotes Landau's big–O notation for asymptotic upper bounds. For comparison, a first-order optimization step has complexity $O(n)$. Moreover, in general case, the practical computational cost of inverting the Hessian matrix is significantly higher, reaching $O(n^3)$. There are three major interesting research directions, that avoid high computational and memory complexity:

**Hessian–free methods** (Martens et al., 2010; Martens & Grosse, 2015b) use the conjugate gradient algorithm to iteratively perform the Newton step (Polyak, 2007) search. The procedure solves for $p$ in $\nabla^2 \ell(\theta)\, p = -\nabla \ell(\theta)$ ($\theta$ refers to model parameters and $\ell$ is a loss function). This requires only Hessian–vector products, computed in $O(n)$ time each, and thus avoiding the explicit $O(n^3)$ inver-

sion of the Hessian. In practice, a small number of conjugate gradient iterations is often sufficient to obtain an adequate search direction.

**L–BFGS variants** (Zocco & McLoone, 2020; Byrd et al., 1994) approximate the Newton step by constructing a low-rank inverse Hessian estimate from a constant number $m$ of recent parameter and gradient differences. Storing these $m$ pairs requires $2mn$ floating–point numbers, which can become a considerable limitation for deep models with very large $n$.

**Approaches utilizing the empirical Fisher information matrix** (empirical Fisher; eFIM), a practical, dataset–based approximation of the true Fisher Information Matrix (FIM). The FIM quantifies curvature of the log-likelihood.

Let $\mathcal{D} = \{(u_i, v_i)\}_{i=1}^N$ be a supervised dataset. We define the per-example negative log-likelihood (NLL) as $\ell(u_i, v_i \mid \theta) := -\log p_\theta(v_i \mid u_i)$, where $v_i$ is the ground-truth class index. For classification with one-hot targets, this is equivalent to the standard cross-entropy loss. The population ("true") FIM is the expected outer product of the score (the gradient of the log-likelihood), with expectations taken over the data-generating distribution $q(u)$ and the model's predictive distribution $p_\theta(\cdot \mid u)$:

$$\boldsymbol{F}(\theta) = \mathbb{E}_{u \sim q(u)} \, \mathbb{E}_{\hat{v} \sim p_\theta(\cdot \mid u)} \Big[ \big( \nabla_\theta \log p_\theta(\hat{v} \mid u) \big) \big( \nabla_\theta \log p_\theta(\hat{v} \mid u) \big)^\top \Big]. \tag{12}$$

Equivalently, since $\nabla_\theta \ell(u, \hat{v} \mid \theta) = -\nabla_\theta \log p_\theta(\hat{v} \mid u)$, we can write

$$\boldsymbol{F}(\theta) = \mathbb{E}_{u \sim q(u)} \, \mathbb{E}_{\hat{v} \sim p_\theta(\cdot \mid u)} \Big[ \big( \nabla_\theta \ell(u, \hat{v} \mid \theta) \big) \big( \nabla_\theta \ell(u, \hat{v} \mid \theta) \big)^\top \Big]. \tag{13}$$

Under standard regularity conditions (support of $p_\theta(v \mid u)$ independent of $\theta$; $p_\theta(v \mid u) > 0$ on its support; and interchange of differentiation and expectation), this equals the expected Hessian of the NLL:

$$\boldsymbol{F}(\theta) = \mathbb{E}_{u \sim q(u)} \, \mathbb{E}_{\hat{v} \sim p_\theta(\cdot \mid u)} \big[ \nabla_\theta^2 \ell(u, \hat{v} \mid \theta) \big]. \tag{14}$$

The **empirical Fisher** is a widely used approximation of Eq. equation **??**. It replaces the expectation over $q(u)$ with a sample average over $\mathcal{D}$, and replaces the inner expectation over $p_\theta(\cdot \mid u)$ with the observed ground-truth labels:

$$\hat{\boldsymbol{F}}(\theta) := \frac{1}{N} \sum_{i=1}^N \big( \nabla_\theta \ell(u_i, v_i \mid \theta) \big) \big( \nabla_\theta \ell(u_i, v_i \mid \theta) \big)^\top. \tag{15}$$

The eFIM is symmetric and positive semidefinite by construction. For stable inversion, it is often damped as $\hat{\boldsymbol{F}}(\theta) + \lambda \boldsymbol{I}$, where $\lambda > 0$.

Crucially, Eqs. 13 and 14 show that the expected Hessian can be obtained efficiently from the expected outer product of gradients, allowing fast estimation of curvature information using Eq. 15 without explicitly computing second derivatives.

In practice, $\hat{\boldsymbol{F}}(\theta)$ is used to precondition gradients in natural-gradient methods (Amari, 1998), with scalable implementations via structured factorizations such as Kronecker-factored blocks (Martens & Grosse, 2015a). Gradient "correlations" can be tracked cumulatively across training iterations $(u^{(t)}, v^{(t)})$ drawn from the training stream that approximates $q$, as in Shampoo (Gupta et al., 2018) (Eq. 15 for $n$ equal to the iteration number) or via an exponential moving average (EMA) that emphasizes recent curvature, as in K-FAC (Grosse & Martens, 2016):

$$M_t = \beta M_{t-1} + (1 - \beta) \big( \nabla_\theta \ell(u^{(t)}, v^{(t)} \mid \theta_t) \big) \big( \nabla_\theta \ell(u^{(t)}, v^{(t)} \mid \theta_t) \big)^\top, \quad \beta \in [0, 1).$$

At the population level, the natural–gradient update direction (Amari, 1998) is:

$$\Delta \theta \; \propto \; \boldsymbol{F}(\theta)^{-1} \, \nabla_\theta L(\theta), \tag{16}$$

where $L(\theta) \coloneqq \mathbb{E}_{(u,v) \sim q(u,v)}\big[\ell(u, v \mid \theta)\big]$.

Moreover, the natural gradient can also be derived from a trust-region formulation. In supervised learning (loss minimization), consider

$$
\min_{\Delta\theta} L(\theta + \Delta\theta) \approx \min_{\Delta\theta} \big(L(\theta) + \nabla_\theta L(\theta)^\top \Delta\theta\big)
$$

$$
= \min_{\Delta\theta} \nabla_\theta L(\theta)^\top \Delta\theta \quad \text{subject to} \quad \tfrac{1}{2}\,\Delta\theta^\top \boldsymbol{F}(\theta)\,\Delta\theta \le \varepsilon, \tag{17}
$$

whose solution is the natural-gradient direction with a step size chosen to satisfy the constraint:

$$
\Delta\theta^\star = -\,\alpha\,\boldsymbol{F}(\theta)^{-1}\nabla_\theta L(\theta), \qquad \alpha = \sqrt{\frac{2\varepsilon}{\nabla_\theta L(\theta)^\top\,\boldsymbol{F}(\theta)^{-1}\,\nabla_\theta L(\theta)}}.
$$

Here, the quadratic form $\tfrac{1}{2}\Delta\theta^\top \boldsymbol{F}(\theta)\Delta\theta$ is the second-order expansion of the KL divergence between the model at $\theta$ and $\theta + \Delta\theta$:

$$
\mathrm{D}_{\mathrm{KL}}\big(p_\theta(\cdot \mid u)\,\big\|\,p_{\theta+\Delta\theta}(\cdot \mid u)\big) \approx \tfrac{1}{2}\,\Delta\theta^\top \boldsymbol{F}(\theta)\,\Delta\theta \quad \text{(averaged over } u \sim q(u)).
$$

Thus, the trust-region radius $\varepsilon$ enforces a bound on the local KL change, so each update corresponds to a fixed-length step (on average over $u \sim q(u)$) in the space of predictive distributions $p_\theta(\cdot \mid u)$, independent of the particular parameterization of the model.

The Fisher $F$ can be efficiently approximated by tracking "gradient correlations" restricted to the layer level via Kronecker-factored blocks, substantially reducing memory and computation (Martens & Grosse, 2015a). These ideas are instantiated in K–FAC (Grosse & Martens, 2016; Martens & Grosse, 2015a), which has demonstrated improved sample efficiency and, in some settings, faster time-to-target than stochastic gradient descent optimizer (Zhang et al., 2023; Vyas et al., 2024). A related approach, Shampoo (Gupta et al., 2018), applies structured, per-mode preconditioners to tensor-shaped parameters, achieving strong sample-efficiency gains compared to first-order optimizers such as Adam (Kingma & Ba, 2014), Adagrad (Duchi et al., 2011), and SGD (Ketkar, 2017), with competitive throughput on ResNet (He et al., 2016), Inception (Szegedy et al., 2015), and Transformer architectures (Vaswani et al., 2017). Building on this line of work, the SOAP optimizer (Vyas et al., 2024) extends the Shampoo algorithm (Gupta et al., 2018) by integrating curvature–aware momentum, thereby enhancing its adaptability to diverse curvature geometries. SOAP reports further improvements in sample efficiency and generalization on large language models, with faster time-to-target than Adam (Kingma & Ba, 2014) in reported experiments.

A key challenge for second-order optimizers is the computational cost of inverting the approximate eFIM. Methods like K–FAC (Grosse & Martens, 2016), Shampoo (Gupta et al., 2018), and SOAP (Vyas et al., 2024) tackle this by using structured approximations. More recently, Eva (Zhang et al., 2023) introduced a more compact eFIM representation and leverages the Sherman–Morrison formula (Sherman & Morrison, 1950) to accelerate the matrix inversion, thereby avoiding the costly eigenvalue decomposition required by SOAP (Vyas et al., 2024). Although the asymptotic ratio between the bottleneck operation's complexity (inversion or decomposition) and the compressed eFIM's size is similar across these methods (Grosse & Martens, 2016; Gupta et al., 2018; Vyas et al., 2024; Zhang et al., 2023), Eva has demonstrated significantly shorter wall–clock training times than both SGD (Ketkar, 2017) and other second–order optimizers (Grosse & Martens, 2016; Gupta et al., 2018). Notably, these speedups are achieved while matching the generalization performance of K-FAC and Shampoo on a range of computer vision benchmarks.

Within the scope of our review, the second–order methods that appear most promising based on reported wall–clock efficiency and generalization performance are SOAP (Vyas et al., 2024) and Eva (Zhang et al., 2023). SOAP has been reported to deliver substantial wall–clock speedups with strong generalization on large language models (LLMs) (Vyas et al., 2024). Eva (Zhang et al., 2023) is a lightweight method that maintains competitive generalization and strong sample–efficiency, as demonstrated on computer–vision benchmarks, while exhibiting asymptotically favorable computational and memory scaling that is comparable, up to constant factors, to first–order optimizers.

## J  DETAILS ON EXPERIMENT SETTINGS

Across all experiments, we set some common generally well–performing optimizer hyperparameters (see Table 8).

Table 8: *Optimizer hyperparameters, that are common across experiments.* $p$ denotes number of batches, every which SOAP computes preconditioning matrices.

| Optimizer | Hyperparameters |
|---|---|
| SOAP | $\beta_1 = 0.95, \beta_2 = 0.95, \beta_1 = 0.95, \beta_2 = 0.95, p = 10$ |
| SOAP AG–3 | $d = 300, \beta_1 = 0.95, \beta_2 = 0.95, \beta_1 = 0.95, \beta_2 = 0.95, p = 10$ |
| Adam | $\beta_1 = 0.9, \beta_2 = 0.999$ |
| Adam AG–3 | $d = 45, \beta_1 = 0.9, \beta_2 = 0.999$ |
| RMSProp | $\beta = 0.99$ |
| RMSProp AG–3 | $d = 45, \beta = 0.99$ |

We used cross–entropy loss, which aligns with theoretical foundations of eFIM used by SOAP (Vyas et al., 2024), and the batch size of 128 across all experiments, except for ResNet–152, where we decreased batch size to 64. We used ResNet–152, which was pretrained on ImageNet–1K dataset (Deng et al., 2009), and we mapped classes to Imagenet–OOD (Yang et al., 2023) to have about 18% accuracy before beginning the fine–tuning.

We omitted some training runs involving optimizers with momentum, like Adam, and forward–looking updates, AG–1 and AG–2. In this scenarios AG–2 modifies a small fraction of an actual update, that is not related to the momentum, limiting the potential of forward–looking concept. AG–1 might negate momentum direction, which also is undesired.

To determine, whether the performance is highly dependent on activation functions, we performed additional experiments after changing ReLU to different activation functions.

## K    DISCLOSURE OF LARGE LANGUAGE MODEL ASSISTANCE

We disclose and delimit the roles played by large language models (LLMs) in the preparation of this manuscript. All LLM–assisted outputs were reviewed and edited by the authors; technical content and decisions remain authorial.

LLMs were used to generate and refine sentence–level phrasing and for localized rewording of paragraphs to improve clarity and concision. AI assistance aided in correcting grammar, improving readability, and harmonizing tone across sections while preserving technical content and author intent. LLMs were consulted to flag potential inconsistencies in notation and equation formatting; all mathematical derivations, identities, and proofs were authored, verified, and finalized by the authors. All LLM–suggested text and equation–formatting changes were reviewed line–by–line by the authors.

LLMs assisted in identifying a minority of potentially relevant papers that did not appear in our initial manual queries. AI tools were used to obtain brief summaries and to triage candidate works for subsequent human evaluation. Final inclusion decisions and categorization were made by the authors after reading the original papers. Citations, summaries, and positioning in the related–work section are based on the original sources. LLM outputs were used only to broaden the candidate set and prioritize reading order.

LLMs were not used to run, tune, or select experiments, nor to generate tables or figures beyond minor wording edits in captions.

