# OpenReview forum: "Efficient Parameter-Space Integrated Gradients for Deep Network Optimization"
_ICLR.cc/2026/Conference — ICLR 2026 Conference Withdrawn Submission_

### Official Review · Reviewer_PxFS · 2025-10-28

**Soundness:** 2
**Presentation:** 1
**Contribution:** 2
**Rating:** 2
**Confidence:** 4

**Summary:**

The authors propose a novel optimizer to train ANNs. Instead of calculating the gradient only at single points in the parameter space, the authors propose to integrate the gradient along a linear path between two parameter points. This gradient integration is approximated by individually integrating each factor of the backpropagation algorithm and assuming a linear dependence of each layer's activations on the parameters. The authors evaluate their proposed method on custom CNNs on the MNIST and Fashion-MNIST dataset as well as on ResNet-152 on Imagenet-OOD.

**Strengths:**

The core concept of gradient integration is interesting.

**Weaknesses:**

1. The paper is difficult to understand in many sections. The mathematical formalism is imprecise and is hard to follow. The lengthy discussion at the paper's conclusion doesn't help much to understand the experimental setup or results.
2. The motivation behind the method is unclear. What is the intuitive advantage of this approach compared to training with momentum? If the gradient is averaged over multiple parameter positions, why not also vary the data batch, similar to momentum? I fail to see an intuitive reason why the proposed method should outperform standard momentum-based training.
3. The specific approximations and assumptions made during the method's derivation, along with their justifications, need to be stated more clearly. For instance, factorizing the gradient averaging (Eq. 2) seems like a very harsh approximation to me that requires more thorough justification.
4. The empirical evaluation is insufficient. The chosen metrics (relative batch-loss minimization and relative sample efficiency) appear contrived, and their practical significance is not evident. Simply demonstrating improvement on these self-defined metrics is not enough to prove the advantages of the proposed method.
5. The authors do not provide  a comparison of the computational runtime of their proposed optimizer against baseline optimizers.
6. Tab. 2-4 are very messy and difficult to interpret. The criteria for bolding numbers are not explained. The meaning or significance of the "step length" column is unclear.
7. The training procedure for Fig. 1 is not described. Imagenet-OOD is not a well-established benchmark. You should give more details what you are doing here.
8. The authors claim their method improves generalization but only provide test accuracies for a single model (ResNet-152) in Figure 1, which show only marginal gains. More extensive experiments on diverse and larger datasets are necessary. While experiments using models like ViT on ImageNet would be ideal, results on tasks such as WideResNet on CIFAR100 should be included at a minimum.
9. If the authors claim the reach better generalization than e.g. SAM, they should directly compare against this method.
10. The paper lacks any theoretical analysis of the proposed method.
11. The attached code is very messy and extremely difficult to comprehend.

**Questions:**

1. How did you choose the model for the LR-optimization in Tab. 2 and 3?
2. What is the exact improvement over your previous work?

---

> ### Author Response · Authors · 2025-11-24
> **Response**
>
> Thank you for your review.
>
> 1. Table 2 presents the results for Model B, with learning rates optimized specifically for that architecture. Similarly, Table 3 displays the results for Model C, where learning rates were tuned for the specific training runs of that model. We selected these models because they exhibit pronounced optimization challenges, such as the Shattered Gradients problem [1] or singularity issues [2].
>
> 2. In this work, we introduce several new variants, AG-2, AG-1 Linear, AG-2 Linear, and AG-3, and extend our evaluation to the Adam and Soap optimizers, whereas our previous work focused only on RMSProp. These expansions resulted in broad performance improvements, most notably a significant gain in test accuracy for ResNet-152 fine-tuning compared to second-order optimization.
>
> [1] David Balduzzi, Marcus Frean, Lennox Leary, JP Lewis, Kurt Wan-Duo Ma, and Brian
> McWilliams. The shattered gradients problem: If resnets are the answer, then what is the
> question? In International conference on machine learning, pp. 342–350. PMLR, 2017.
> doi:10.48550/arXiv.1702.08591.
>
> [2] Oyebade Oyedotun, Kassem Al Ismaeil, and Djamila Aouada. Training very deep neural networks: Rethinking the role of skip connections. Neurocomputing, 441, 02 2021.
> doi:10.1016/j.neucom.2021.02.004.
>
>
> Response to Weaknesses:
> 1. We would appreciate it if you could specify which mathematical expressions appeared imprecise so that we can address them directly.
>
> 2. [Momentum explanation retained as above]
>
> 3. The justification for factorizing the gradient averaging is provided in Appendix B and the reference cited near the equation.
>
> 4. Relative sample efficiency compares the required training iterations, a standard evaluation method. Accuracy is also a standard metric. Relative batch-loss minimization corresponds to direct loss minimization in full-batch scenarios (e.g., logistic regression or few-shot fine-tuning). All other metrics, including full loss curves, are in the appendices.
>
> 5. "Step length" refers to the mean distance of the optimizer's update step. We report both Manhattan and Euclidean distances, as specified in the paper.
>
> 6. Bold values indicate an improvement over the respective baseline optimizer (e.g., Adam AG-2 vs. Adam).
>
> 7. Training procedures are detailed in the Fig. 1 subcaptions, and AG-3 is defined in Algorithm 1. We used ImageNet-OOD to test robustness to distribution shifts. Our goal was to compare optimization methods relative to one another rather than chasing SOTA results on this specific benchmark.
>
> 8. Although the paper focuses on the broad applicability of integrating gradients, we demonstrate generalization improvements for Model B (Appendix G) and ResNet-152 using Soap, RMSProp, and Adam.
>
> 9. We explicitly limited our comparison with SAM to computational efficiency and parallelizability; we do not claim superior generalization performance. Consequently, empirical generalization experiments were unnecessary for this specific point. Instead, we rely on established complexity analysis: it is well-known that SAM requires two forward and backward passes per update, whereas AG-3 avoids this double-computation overhead by performing a single backward pass.
>
> 10. For the theoretical analysis, please refer to our related manuscript, "Towards enhancing RMSProp..." (in review). A copy of this work has been provided within the supplementary material of this submission.
>
> 11. We acknowledge that the submitted code is currently in a research-prototype state, heavily instrumented for the extensive analysis and logging described in the paper. We are committed to refactoring, commenting, and cleaning the codebase to ensure high readability and reproducibility for the camera-ready version. In the meantime, the exact logical flow of the method is rigorously defined in Algorithms 1 and 2 in the main text.

---

> > ### Comment · Reviewer_PxFS · 2025-11-25
> >
> > Thank you for your response. However, my primary concern, the insufficient empirical evaluation, remains unaddressed. The reported experiments do not convincingly demonstrate improved generalization or computational efficiency on meaningful datasets or models. I will therefore keep my original score.

---

### Official Review · Reviewer_3x66 · 2025-10-31

**Soundness:** 1
**Presentation:** 1
**Contribution:** 2
**Rating:** 2
**Confidence:** 4

**Summary:**

The paper proposes an optimization approach that replaces the standard gradient update with an integrated gradient. Instead of using the instantaneous gradient, the method averages gradients computed along a virtual path between two or more weight states, claiming this approximates multiple gradient steps in one update. The paper introduced several variants, corresponding to integrating over one, two, or three previous parameter states. Experiments on small CNNs and ResNet-152 fine-tuning show lower per-batch losses and faster training compared to SGD and Adam, based on a custom metric called “Relative Batch–Loss Minimization.”

**Strengths:**

The paper may contain an interesting idea, but critical details are omitted or deferred to a non-public (non-peer-reviewed) reference in the supplementary material, making it difficult to judge the technical contribution.

**Weaknesses:**

The paper is poorly written and difficult to evaluate. The introduction fails to clearly describe the problem, motivation, or context, and key details are deferred to an anonymized, non-public paper that is cited as “under review” in the supplementary material. This is unprofessional and makes an assessment of the claimed contributions difficult. The paper frequently references prior “results” without providing credible sources other than a different manuscript in the supplementary material, leaving the reviewer unable to determine what is being extended or improved upon.

Technically, the proposed method, described as an “integrated gradient in parameter space”, reduces to a simple averaging of current and past gradients, offering no meaningful novelty beyond classical momentum or lookahead mechanisms. The presentation lacks reasoning, explanations, and substantive discussion. The writings are difficult to follow and overloaded with vague or loosely defined terms.

The experiments are limited to CNNs, and there is no clear reason why the proposed method can not be tested on different architectures on more standard benchmarks and tasks. The paper reports results using an unconventional metric (“Relative Batch Loss Minimization”) that does not align with standard evaluation practices. The overall formatting and clarity are poor, and the work does not meet the standards for a publishable submission.

**Questions:**

Beyond implementation details, what is the fundamental conceptual or algorithmic difference between the proposed “integrated gradient” update and classical momentum or lookahead methods, which also average gradients across iterations?

---

> ### Author Response · Authors · 2025-11-24
> **Response**
>
> Our variants, such as Soap AG-3, incorporate momentum and empirically outperform standard Soap with momentum, demonstrating that our method operates beyond classical momentum mechanics.
>
> Consider that classical momentum is effectively influenced by $n$ gradients from previous updates (defined by a decay threshold), where each gradient is computed on different parameter states and batches. In contrast, with our method, each of these $n$ contributions is derived using weights from the last $k$ training iterations ($k$ being our hyperparameter).
>
> In summary, each iteration of our method utilizes information effectively equivalent to $n \cdot k$ gradients, compared to just $n$ gradients in classical momentum. For Soap AG-3, where $k$ ranges from 300 to 600, our method aggregates information from at least $300n$ gradients per update. This significantly enhances the detection of sharp local minima and improves generalization. Crucially, we employ efficient formulas to estimate these aggregated gradients immediately without computational overhead.
>
>
> Response to Weaknesses:
> 1. We have provided the manuscript of our previous work to clearly demonstrate the novelty and distinct contributions of the current submission in comparison to our prior findings.
>
> 2. As detailed in the "Theoretical Connection to Momentum" section above, our method fundamentally differs from classical momentum by utilizing a history of weight updates ($n \cdot k$ gradients) rather than a simple decaying average of past gradients.
>
> 3. Convolutional Neural Networks (CNNs) remain the primary architecture in many Deep Reinforcement Learning scenarios.
>
> In specific cases, the training batch effectively equals the whole dataset, for instance, in large-scale gradient-based logistic regression or when fine-tuning a neural network with a limited number of examples (few-shot learning). In such full-batch scenarios, "Relative Batch Loss Minimization" becomes mathematically equivalent to standard global loss minimization.

---

### Official Review · Reviewer_PEvM · 2025-11-01

**Soundness:** 2
**Presentation:** 3
**Contribution:** 2
**Rating:** 4
**Confidence:** 3

**Summary:**

This paper proposes a novel and efficient method for optimizing deep neural networks by using integrated gradients in parameter space. Instead of relying on the instantaneous gradient at the current parameters, the core idea is to compute an averaged gradient over a continuous path between two parameter points (e.g., from past to current, or current to a forecasted future state) for each update.

**Strengths:**

The experimental results are good. AG-1 and AG-2 achieve up to 53.5% greater per-batch loss reduction compared to baseline optimizers at matched step lengths. AG-3 provides consistent test accuracy gains (~0.4%) on ResNet-152 fine-tuning, a notable improvement over a strong second-order baseline.

**Weaknesses:**

1. In section 3.2 the authors show the algorithm is based on the approximation and propagation of the average gradient according to  the paper [1], which is under review and cannot be found now. I believe this will have a significantly negative impact on the reviewers' assessment of this paper's contributions.

2. The need to modify the backpropagation graph for each layer type makes it non-trivial to port to new architectures.

3. The comparison with the baseline of the Transformer-based model is missing.

**Questions:**

It is unclear why the performance reported in Figure 1 remains below 60% for all methods. Additionally, the observed decline in accuracy before training reaches 10 epochs warrants clarification.

---

> ### Author Response · Authors · 2025-11-24
> **Response**
>
> Thank you for your review. The performance shown in Fig. 1 is below 60% because we utilized ImageNet-OOD [1], a smaller version of ImageNet. This dataset averages approximately 50 examples per class, whereas the standard ImageNet-1K contains over 1,000 images per class (on average).
>
> While ResNet-152 achieved 78.57% accuracy on ImageNet-1K in the original paper [2], we anticipated a drop to below 60% given the significant reduction in training examples.
>
> Furthermore, due to the limited dataset size, the model suffered from overfitting after only a few epochs of fine-tuning, resulting in the observed decline in accuracy.
>
> [1] William Yang, Byron Zhang, and Olga Russakovsky. Imagenet-ood: Deciphering modern out-of-distribution detection algorithms. arXiv preprint arXiv:2310.01755, 2023.
> doi:10.48550/arXiv.2310.01755.
> [2] He, K., Zhang, X., Ren, S., & Sun, J. (2016). Deep residual learning for image recognition. In Proceedings of the IEEE conference on computer vision and pattern recognition (pp. 770-778).

---

### Official Review · Reviewer_wCxi · 2025-11-01

**Soundness:** 2
**Presentation:** 2
**Contribution:** 1
**Rating:** 2
**Confidence:** 3

**Summary:**

The paper proposes an optimization framework called **Averaged Gradient (AG)** or **parameter-space integrated gradients**, which modifies backpropagation by replacing the instantaneous gradient with its average along a short trajectory in parameter space.  Unlike conventional momentum methods that aggregate past gradients with exponential weights, AG integrates the gradient field continuously (or approximately) between two or more parameter states, requiring no additional hyperparameters.  Several variants—AG-1 (forward-sign), AG-2 (full-magnitude), AG-3 (multi-step backward averaging), and AG-3-1 (one-step variant)—are introduced and evaluated across CNN, fully-connected, and ResNet architectures using optimizers such as Adam, RMSProp, and SOAP.  Experiments show that AG yields up to 50% faster batch-loss minimization and modest but consistent generalization gains, approximating the effect of multiple gradient updates at only twice the computational cost of a standard backward pass.

**Strengths:**

The topic of integrating gradients along a path in parameter space is conceptually important and connects to a fundamental idea in both optimization and interpretability.  The paper provides a computationally efficient extension of integrated gradients to the context of network training, showing that parameter-space integration can improve loss minimization and generalization with minimal overhead.

**Weaknesses:**

1. **Lack of technical contribution.** The core idea of integrated gradients is well-established, and the key innovation—the Averaged Gradient (AG) algorithm—was already introduced in the cited anonymous work. This paper mainly provides implementation variants (AG-2, AG-3, etc.) and additional empirical studies, rather than new theoretical or methodological advances.
2. **Obscure writing and weak presentation.** Many essential definitions and explanations (e.g., the definition of the averaging operator in Eq. (2)) are deferred to the anonymous citation or the appendix, making the paper hard to follow. The layout also has problems: some parts are overly dense (Table 1), while others are too sparse (Figure 1) or extend beyond column width (e.g., line 173). The overall presentation requires substantial revision before publication.
3. **Limited experimental scale.** The ResNet-152 evaluation is limited to fine-tuning rather than full end-to-end training, leaving open whether AG methods remain practical for large-scale optimization. The smaller models (A, B, C) contain only about 0.01 M parameters, which are too simple to convincingly demonstrate the claimed efficiency benefits.
4. **Relative reporting of results.** All numerical results in Tables 2–4 are given only as relative improvements, obscuring the absolute performance and making it difficult to assess convergence behavior or generalization. It is also unclear whether the baseline optimizers were sufficiently tuned, which could exaggerate the apparent advantages of the AG variants.

**Questions:**

1. Intuitively, the gradient at the current update should have greater significance. In previous methods such as momentum, there are hyperparameters to control this weighting, but in AG methods the integration is uniform with no weights. Would introducing weighted integration improve performance or stability?
2. Could you provide a direct quantitative comparison of time and memory consumption between the AG methods and conventional optimizers (e.g., Adam, SAM, or SOAP) to clarify the computational trade-offs?

---

> ### Author Response · Authors · 2025-11-24
> **Response**
>
> Thank you for your review.
>
> 1. Based on our intuition, weighted integration might improve performance or stability in certain use cases, though not in all. For example, switching to weighted integration instead of the standard one in Eq. 1 would invalidate the approximated equality between $n$ weight updates and the integral of gradients, leading to the impossibility of approximating multiple weight updates using the average gradient.
> Secondly, our formula for the efficient computation of the approximated integral of gradients is not applicable to weighted integration. Thus, weighted integration might be significantly less computationally efficient.
>
> 2. Runtime impact varies by method variant. We did not include runtime plots in the paper because our implementation prioritizes analysis over speed; it includes heavy instrumentation (such as inference steps after every update, or a separate pass through all training examples to have an accurate training loss at the end of each epoch) and lacks optimization.
>
> Given this context, the direct runtime results for ResNet-152 on ImageNet-OOD are as follows:
> - Adam vs. Adam AG-3: Using Adam AG-3, 100 training runs took 2 days, 16 hours, and 5 minutes ($\approx$ 38.5 min/run). In comparison, standard Adam completed 200 training runs in 1 day, 22 hours, and 33 minutes ($\approx$ 14.0 min/run). This indicates a runtime factor of $\approx$2.75$\times$ for this unoptimized setup.
> - Soap vs. Soap AG-3: For the Soap variants (preconditioned every batch), Soap AG-3 took 2 days, 15 hours, and 56 minutes for 50 runs ($\approx$ 76.7 min/run), whereas standard Soap took 3 days, 7 hours, and 20 minutes for 75 runs ($\approx$ 63.5 min/run). Here, our method is only $\approx$21% slower, as the second-order preconditioning is the computational bottleneck.
>
> At present, we do not have statistics on allocated GPU memory or the empirical computational efficiency of SAM. We would need additional time to conduct the necessary SAM training runs and GPU allocation profiling.
>
>
>
> Response to Weaknesses:
> 1. To the best of our knowledge, no optimization method utilizing integrated gradients has previously been proposed in peer-reviewed literature. Therefore, this represents a novel contribution rather than an iteration on well-established methods.
>
> 2. To adhere to space constraints and improve flow, we moved standard definitions and supplementary details to the appendices.
>
> 3. We intentionally limited the experimental scale to facilitate robustness; this allowed us to perform hundreds of training runs per model and optimization method, ensuring rigorous hyperparameter tuning.
>
> 4. The absolute values for the metrics in Tables 2–4 are provided in the appendices in the form of learning curves. This allows for a detailed assessment of convergence behavior and generalization.

---

### Note · Authors · 2026-01-23

**Comment:**

We have decided to withdraw our paper from ICLR 2026. We would like to thank the reviewers for their constructive comments and the Area Chair for overseeing the review process. We will use the feedback to improve our work for future submissions.

**Withdrawal Confirmation:**

I have read and agree with the venue's withdrawal policy on behalf of myself and my co-authors.